# LiteXrayNet: Bilateral Asymmetry-Aware Attention for Lightweight Pediatric Pneumonia Detection

## Abstract

Pediatric pneumonia remains a major cause of mortality among children under five, with the greatest burden in resource-constrained settings where access to timely diagnosis is limited. Although deep learning methods have achieved strong performance in chest X-ray analysis, many existing approaches rely on large models that are difficult to deploy in such environments and do not explicitly account for the bilateral anatomical structure that radiologists routinely use during interpretation. We present LiteXrayNet, a lightweight convolutional neural network that incorporates Bilateral Asymmetry Attention (BAA), a geometry-guided attention mechanism designed to model left-right lung correspondence through spatial splitting, horizontal flipping, and adaptive feature gating. With only 127K parameters, LiteXrayNet achieves competitive pneumonia classification performance, attaining an F1 score of 97.31% and an accuracy of 97.90%, while supporting real-time inference on edge hardware with latencies of $4.11\,\mathrm{ms}$ on GPU and $14.53\,\mathrm{ms}$ on CPU. Feature-level bilateral asymmetry analysis indicates that BAA induces representations that differ systematically from those produced by generic attention mechanisms, while Grad-CAM visualizations suggest anatomically structured attention patterns consistent with common radiological reasoning. These results suggest that incorporating domain-specific anatomical priors as architectural constraints can support efficient and interpretable models suitable for deployment in resource-limited clinical settings.

## 1 Introduction

Pneumonia remains one of the leading causes of mortality among children under five years of age, accounting for approximately 14% of deaths in this group and claiming over 740,000 lives annually World Health Organization (2023). The burden is greatest in resource-constrained regions, where access to timely diagnosis and specialized radiological expertise is often limited. Chest X-ray imaging is the primary diagnostic modality, yet accurate interpretation requires trained radiologists who routinely compare bilateral lung fields to identify pathological patterns.

Deep learning methods have achieved strong performance in automated pneumonia detection from chest X-rays Rajpurkar et al. (2017); Kermany et al. (2018). However, many high-performing models rely on large architectures with millions of parameters, limiting their practicality for deployment in low-resource or edge settings. Lightweight alternatives improve efficiency but often do so at the cost of reduced diagnostic accuracy. In addition, commonly used attention mechanisms Hu et al. (2018); Woo et al. (2018); Wang et al. (2020) model spatial dependencies in a generic manner and do not explicitly encode the bilateral anatomical structure that is central to radiological interpretation.

In clinical practice, radiologists systematically assess left and right lung fields, as diagnosis depends on bilateral comparison even when disease manifestations are asymmetric or diffuse. Prior work has shown that incorporating anatomical constraints into neural architectures can improve robustness and generalization in medical imaging tasks Oktay et al. (2018); Ravishankar et al. (2019); Chen et al. (2021). Pediatric chest radiography is particularly well suited for exploiting bilateral structure as an architectural prior, given standardized acquisition protocols and relatively consistent anatomical orientation.

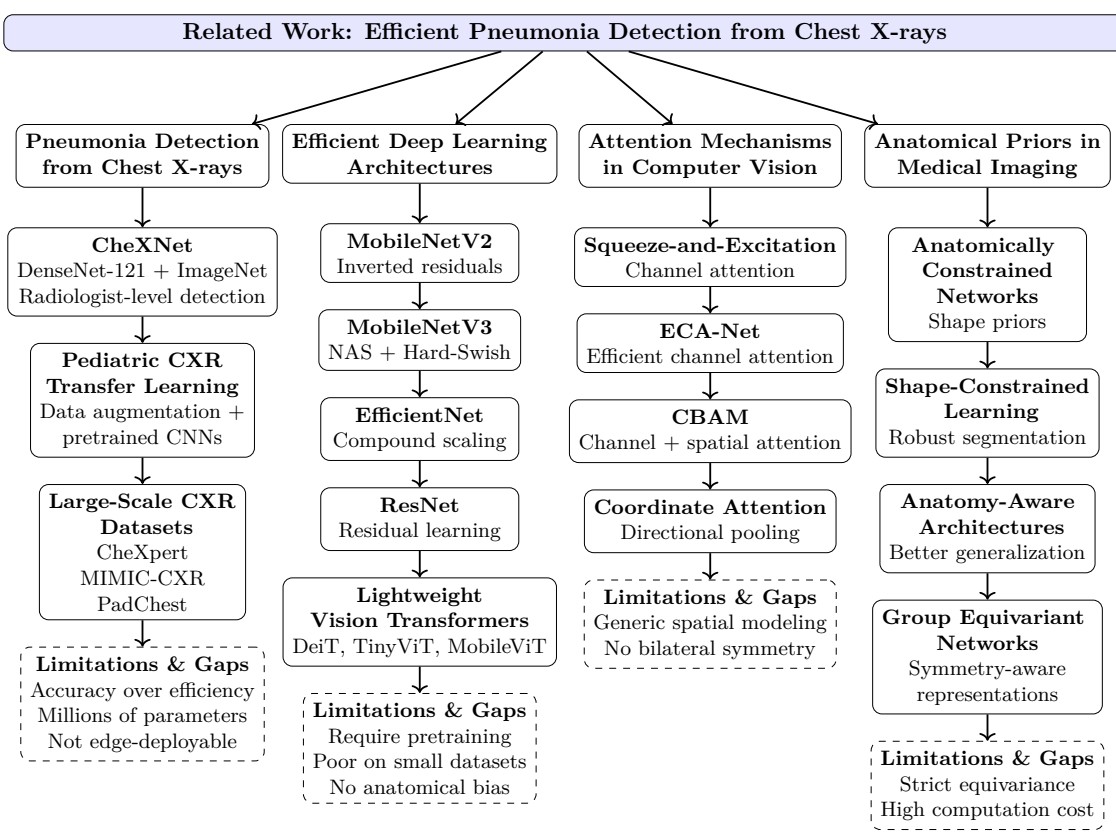

Figure 1: Overview of prior work in pneumonia detection from chest X-rays, efficient architectures, attention mechanisms, anatomical priors, and interpretability, highlighting their limitations.

In this work, we propose LiteXrayNet, a lightweight convolutional neural network that incorporates Bilateral Asymmetry Attention (BAA), a geometry-guided attention mechanism designed to model left-right lung correspondence through spatial splitting, horizontal flipping, and learnable feature gating. With approximately 127K parameters, LiteXrayNet achieves competitive pneumonia classification performance while supporting real-time inference on edge hardware. Feature-level bilateral asymmetry analysis indicates that BAA induces representations that differ systematically from those learned by generic attention mechanisms, and qualitative visualizations suggest attention patterns that are more anatomically structured and aligned with common radiological reasoning.

The remainder of this paper is organized as follows. Section 2 reviews related work. Section 3 describes the LiteXrayNet architecture and the BAA formulation. Section 4 outlines the experimental setup. Section 5 presents the evaluation results, and Section 6 discusses findings and limitations. Section 7 concludes the paper.

## 2 Related Work

Deep learning approaches have demonstrated strong performance in pneumonia detection from chest X-rays. CheXNet Rajpurkar et al. (2017) pioneered this direction by achieving radiologist-level performance using ImageNet-pretrained DenseNet architectures. Subsequent work, including that of Kermany et al. Kermany et al. (2018), showed that transfer learning combined with data augmentation can be effective for pediatric chest radiographs. The availability of large-scale datasets such as CheXpert Irvin et al. (2019), MIMIC-CXR Johnson et al. (2019), and PadChest Bustos et al. (2020) has further enabled the development of high-capacity models. However, these approaches typically prioritize predictive performance over computational

efficiency and rely on architectures with millions of parameters, limiting their suitability for deployment in resource-constrained or edge environments.

To address efficiency, a range of lightweight architectures have been proposed. MobileNetV2 Sandler et al. (2018) and MobileNetV3 Howard et al. (2019) reduce computational cost through depthwise separable convolutions, inverted residuals, and neural architecture search. EfficientNet Tan & Le (2019) demonstrated that compound scaling can yield improved accuracy–efficiency trade-offs, while ResNet He et al. (2016) remains a commonly used baseline despite its relatively large parameter count. More recently, lightweight vision transformers such as DeiT Touvron et al. (2021), TinyViT Wu et al. (2022), and MobileViT Mehta & Rastegari (2022) have been explored for medical imaging tasks. While these models offer architectural diversity, they often require extensive pretraining or exhibit reduced stability when trained from scratch on smaller medical datasets, and they do not incorporate task-specific anatomical structure.

Attention mechanisms have been widely adopted to improve feature representations in convolutional networks. Squeeze-and-Excitation networks Hu et al. (2018) introduced channel-wise recalibration through global pooling and gating, while ECA-Net Wang et al. (2020) improved efficiency by replacing fully connected layers with lightweight convolutions. CBAM Woo et al. (2018) combined channel and spatial attention, and Coordinate Attention Hou et al. (2021) encoded directional spatial information through axis-wise pooling. Although effective in general vision tasks, these mechanisms model spatial relationships in a generic manner and do not explicitly encode bilateral correspondence between anatomically mirrored regions, which is central to chest X-ray interpretation.

Several studies have shown that incorporating anatomical priors into neural architectures can improve robustness and generalization in medical imaging. Oktay et al. (2018) introduced anatomically constrained networks using shape priors for cardiac segmentation, while Ravishankar et al. (2019) and Chen et al. (2021) demonstrated that anatomy-aware designs can improve performance across imaging protocols. Group equivariant convolutional networks Cohen & Welling (2016); Weiler & Cesa (2019) provide a principled framework for encoding symmetry but are unsuitable for this application on two grounds. First, equivariant convolutions increase FLOPs by a factor proportional to the group size $|G|$; for the reflection group relevant to bilateral lung symmetry, this results in approximately twice the computational cost of a standard convolution at equivalent channel width Weiler & Cesa (2019), which is incompatible with our edge deployment target. Second, and more fundamentally, strict reflection equivariance forces identical feature representations for left-right mirrored inputs, suppressing the side-specific pathological information — unilateral consolidation, asymmetric effusion — that is diagnostically relevant in chest radiography. Among the generic attention mechanisms evaluated in this work, Coordinate Attention exhibits the most aggressive bilateral symmetrization, achieving a near-zero BAS of 0.007 (Table 5), and correspondingly produces the lowest F1 score among attention variants, providing direct empirical support for this representational argument. In contrast, BAA encodes bilateral correspondence through soft geometric constraints while explicitly preserving side-specific information through a learned asymmetric branch and adaptive gate. Table 1 provides a structured analytical comparison of these approaches across computational and representational dimensions.

Table 1: Analytical comparison of attention and symmetry-aware approaches across criteria relevant to bilateral chest X-ray analysis. BAS values for attention mechanisms are from Table 5. G-CNN FLOPs overhead estimated from published complexity analysis in Weiler & Cesa (2019).

| Method | Strict equivariance | Cross-side coupling | Asymmetric repr. | Edge suitable |
|---|---|---|---|---|
| G-CNN Cohen & Welling (2016) | Yes | Yes | No | No |
| SE-Net Hu et al. (2018) | No | No | Yes | Yes |
| CBAM Woo et al. (2018) | No | No | Yes | Yes |
| Coord. Attn Hou et al. (2021) | No | No | No | Yes |
| BAA (ours) | No | Yes | Yes | Yes |

Interpretability methods such as Grad-CAM Selvaraju et al. (2017) are widely used to visualize class-discriminative regions in medical images, but their outputs require careful interpretation and are best viewed

as complementary to quantitative analysis Doshi-Velez & Kim (2017); Tjoa & Guan (2020); Arun et al. (2021). Prior work has largely focused on qualitative visualization and has not examined whether internal representations systematically reflect anatomical structure. We address this gap by introducing feature-level bilateral asymmetry analysis to evaluate how geometry-guided attention influences learned representations.

## 3 Method

### 3.1 Problem Formulation

Given a chest X-ray image $\mathbf{X} \in \mathbb{R}^{H \times W}$ resized to $224 \times 224$ pixels, pneumonia detection is formulated as a binary classification task $f : \mathbf{X} \to y$, where $y \in \{0, 1\}$ denotes the NORMAL or PNEUMONIA class. The objective is to learn a function $f$ that achieves strong diagnostic performance while maintaining a compact parameter footprint suitable for edge deployment, and that supports anatomically interpretable feature representations. We assume standard clinical preprocessing with approximate image centering and consistent frontal orientation, as is typical for pediatric chest radiographs. No explicit lung segmentation or spatial registration is performed.

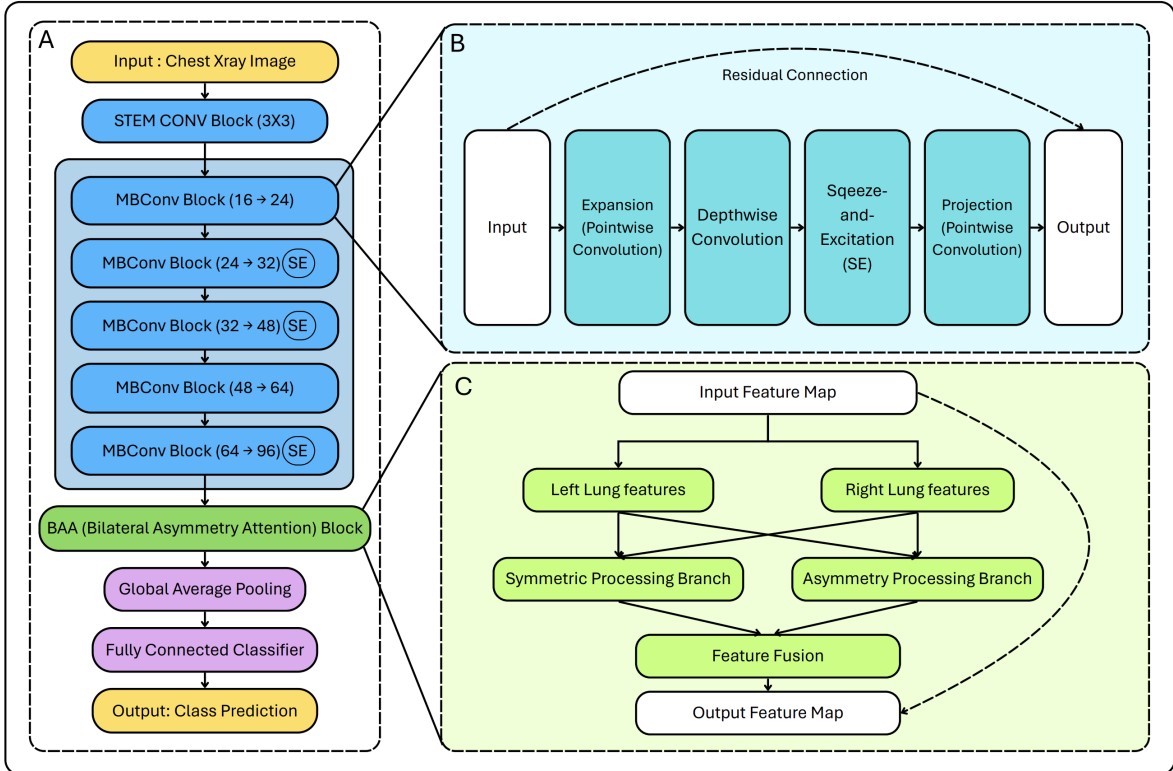

Figure 2: LiteXrayNet architecture overview. **(A)** Complete pipeline showing stem convolution, five MB-Conv blocks with progressive channel expansion, Bilateral Asymmetry Attention (BAA) module, and classifier head. **(B)** MBConv block structure with expansion, depthwise convolution, Squeeze-and-Excitation, projection, and residual connection. **(C)** BAA module mechanism showing spatial splitting into left and right lung features, parallel symmetric and asymmetric processing branches, and adaptive feature fusion with residual connection.

### 3.2 LiteXrayNet Architecture Overview

LiteXrayNet consists of four main components: an initial stem for feature extraction, a backbone composed of efficient MBConv blocks, the proposed Bilateral Asymmetry Attention (BAA) module, and a lightweight classifier head. An overview of the architecture is shown in Figure 2.

The stem comprises a $3 \times 3$ convolution with stride 2, followed by batch normalization and the HSwish activation function, producing 16 feature channels from the input grayscale image. The backbone progressively increases channel dimensionality through five MBConv stages with channel sizes $16 \rightarrow 24 \rightarrow 32 \rightarrow 48 \rightarrow 64 \rightarrow 96$. The BAA module operates on the final 96-channel feature map to introduce explicit bilateral coupling. The classifier head applies global average pooling followed by a two-layer fully connected network with dropout regularization and softmax activation for binary prediction. The complete architecture contains 127,674 trainable parameters.

### 3.3 MBConv Block Design

The MBConv blocks follow the design principles of MobileNetV3 Howard et al. (2019), leveraging depthwise separable convolutions to reduce computational cost. Each block consists of four sequential operations. First, channel expansion is performed using a $1 \times 1$ convolution with an expansion factor of 2 to increase representational capacity. Second, spatial feature extraction is carried out using a $3 \times 3$ depthwise convolution with batch normalization and HSwish activation. Third, a Squeeze-and-Excitation module with a reduction ratio of 4 recalibrates channel responses by modeling inter-channel dependencies via global pooling and gating. Finally, a $1 \times 1$ convolution projects features back to the target output dimensionality. Residual connections are applied when the input and output dimensions are compatible, facilitating gradient flow and feature reuse.

This design provides an efficient backbone suitable for edge deployment while retaining sufficient representational capacity for medical image analysis. The Squeeze-and-Excitation modules offer channel-wise recalibration that complements the spatially focused bilateral modeling introduced by the BAA module, without explicitly encoding left-right correspondence.

### 3.4 Bilateral Asymmetry Attention

#### 3.4.1 Motivation

Radiologists routinely compare left and right lung fields during chest X-ray interpretation, as diagnostic decisions often depend on bilateral assessment regardless of whether disease manifestations are unilateral or diffuse. In contrast, commonly used attention mechanisms such as SE-Net Hu et al. (2018), CBAM Woo et al. (2018), and ECA-Net Wang et al. (2020) model feature dependencies without explicitly coupling anatomically mirrored regions. We posit that exposing bilateral correspondence through simple geometric operations, while allowing the network to learn the relative importance of symmetric and asymmetric patterns, can improve representation quality without enforcing strict equivariance as in group-theoretic approaches Cohen & Welling (2016); Weiler & Cesa (2019).

#### 3.4.2 Mechanism

Algorithm 1 summarizes the complete LiteXrayNet forward pass. The BAA module (Steps 7–15) processes left and right lung regions as follows.

$$\mathbf{F}_{\text{left}} = \mathbf{F}[:, :, :, : W/2], \quad \mathbf{F}_{\text{right}} = \mathbf{F}[:, :, :, W/2 :]. \tag{1}$$

Each half is processed by a shared symmetric feature extractor $\phi_{\text{sym}}$, implemented using depthwise separable convolutions:

$$\mathbf{F}_{\text{left}}^{\text{sym}} = \phi_{\text{sym}}(\mathbf{F}_{\text{left}}), \quad \mathbf{F}_{\text{right}}^{\text{sym}} = \phi_{\text{sym}}(\mathbf{F}_{\text{right}}). \tag{2}$$

---

**Algorithm 1** LiteXrayNet Forward Pass

---

**Require:** Chest X-ray image $X \in \mathbb{R}^{H \times W}$
**Ensure:** Class prediction $\hat{y} \in \{0, 1\}$

***Stage 1: Stem***                                           `[architectural choice]`
1: $F_0 \leftarrow \text{HSwish}(\text{BN}(\text{Conv}_{3 \times 3, s2}(X)))$        $\triangleright$ $X \in \mathbb{R}^{224 \times 224} \to F_0 \in \mathbb{R}^{16 \times 112 \times 112}$

***Stage 2: MBConv Backbone***                                  `[architectural choice]`
2: $F_1 \leftarrow \text{MBConv}(F_0;\ 16 \to 24,\ r=2,\ \text{SE=False})$
3: $F_2 \leftarrow \text{MBConv}(F_1;\ 24 \to 32,\ r=2,\ \text{SE=True})$
4: $F_3 \leftarrow \text{MBConv}(F_2;\ 32 \to 48,\ r=2,\ \text{SE=True})$
5: $F_4 \leftarrow \text{MBConv}(F_3;\ 48 \to 64,\ r=2,\ \text{SE=False})$
6: $F_5 \leftarrow \text{MBConv}(F_4;\ 64 \to 96,\ r=2,\ \text{SE=True})$      $\triangleright$ $F_5 \in \mathbb{R}^{96 \times 14 \times 14}$; SE omitted at low-channel stages

***Stage 3: Bilateral Asymmetry Attention (BAA)***                 `[primary contribution]`
7: $F_{\text{left}}, F_{\text{right}} \leftarrow \text{Split}(F_5,\ W/2)$   $\triangleright$ Eq. 1: midline split; valid under standard frontal acquisition (Sec. 3.1)
8: $F_{\text{left}}^{\text{sym}} \leftarrow \phi_{\text{sym}}(F_{\text{left}}); \quad F_{\text{right}}^{\text{sym}} \leftarrow \phi_{\text{sym}}(F_{\text{right}})$                  $\triangleright$ Eq. 2: shared weights
9: $F_{\text{avg}} \leftarrow (F_{\text{left}}^{\text{sym}} + F_{\text{right}}^{\text{sym}})/2$
10: $F_{\text{symmetric}} \leftarrow [F_{\text{avg}},\ \text{Flip}(F_{\text{avg}})]$          $\triangleright$ Eq. 3: bilateral reconstruction via horizontal flip
11: $F_{\text{asym}} \leftarrow |F_{\text{left}} - \text{Flip}(F_{\text{right}})|$     $\triangleright$ Eq. 4: asymmetry map captures side-specific pathological differences
12: $F_{\text{asymmetric}} \leftarrow [\phi_{\text{asym}}(F_{\text{asym}}),\ \text{Flip}(\phi_{\text{asym}}(F_{\text{asym}}))]$                    $\triangleright$ Eq. 5
13: $\alpha \leftarrow \sigma(\text{Conv}(\text{ReLU}(\text{Conv}(\text{GAP}(F_{\text{asymmetric}})))))) \cdot \tau$ $\triangleright$ Eq. 6: $\tau$ learnable scalar init. 1.0, post-sigmoid for gradient stability
14: $F_{\text{fused}} \leftarrow \text{Conv}_{1 \times 1}([F_{\text{symmetric}} \cdot (1-\alpha),\ F_{\text{asymmetric}} \cdot \alpha])$       $\triangleright$ Eq. 7: adaptive weighted fusion
15: $F_{\text{out}} \leftarrow F_5 + F_{\text{fused}}$          $\triangleright$ Eq. 8: residual connection preserves backbone features

***Stage 4: Classifier Head***                                        `[architectural choice]`
16: $z \leftarrow \text{GAP}(F_{\text{out}})$
17: $z \leftarrow \text{HSwish}(\text{BN}(\text{FC}_{96 \to 256}(z)))$
18: $z \leftarrow \text{Dropout}(z)$
19: $\hat{y} \leftarrow \text{Softmax}(\text{FC}_{256 \to 2}(z))$

**Total parameters:** 127K   |   **Our Contribution:** Steps 7–15 (BAA)
**Note:** LiteXrayNet as a whole is the architectural contribution of this work. The BAA module is the primary novel component; all other stages are deliberate design choices optimized for edge deployment.

---

The resulting features are averaged to form a symmetric representation and reconstructed bilaterally:

$$\mathbf{F}_{\text{avg}} = \frac{\mathbf{F}_{\text{left}}^{\text{sym}} + \mathbf{F}_{\text{right}}^{\text{sym}}}{2}, \quad \mathbf{F}_{\text{symmetric}} = [\mathbf{F}_{\text{avg}}, \text{Flip}(\mathbf{F}_{\text{avg}})]. \tag{3}$$

To capture diagnostically relevant differences between lung fields, an asymmetry map is computed by comparing the left features with a horizontally flipped version of the right features:

$$\mathbf{F}_{\text{asym}} = |\mathbf{F}_{\text{left}} - \text{Flip}(\mathbf{F}_{\text{right}})|. \tag{4}$$

This map is processed by a dedicated asymmetric feature extractor $\phi_{\text{asym}}$ and reconstructed:

$$\mathbf{F}_{\text{asymmetric}} = [\phi_{\text{asym}}(\mathbf{F}_{\text{asym}}), \text{Flip}(\phi_{\text{asym}}(\mathbf{F}_{\text{asym}}))]. \tag{5}$$

A learnable gate controls the relative contribution of symmetric and asymmetric information:

$$\alpha = \sigma(\text{Conv}_{1 \times 1}(\text{ReLU}(\text{Conv}_{1 \times 1}(\text{GAP}(\mathbf{F}_{\text{asymmetric}}))))) \cdot \tau, \tag{6}$$

where $\tau$ is a learnable temperature parameter and $\sigma$ denotes the sigmoid function. The two branches are fused and added to the original features via a residual connection:

$$\mathbf{F}_{\text{fused}} = \text{Conv}_{1 \times 1}([\mathbf{F}_{\text{symmetric}} \cdot (1 - \alpha), \mathbf{F}_{\text{asymmetric}} \cdot \alpha]), \tag{7}$$

$$\mathbf{F}_{\text{out}} = \mathbf{F} + \mathbf{F}_{\text{fused}}. \tag{8}$$

### 3.4.3 Design Principles

BAA combines explicit geometric operations—spatial splitting, horizontal flipping, and feature differencing—with learnable gating to expose bilateral structure while retaining flexibility. Symmetry is encouraged through shared weights across lung regions rather than strict pixel-wise constraints, allowing tolerance to natural anatomical variation and minor misalignment. The separation of symmetric and asymmetric processing enables the model to adaptively balance shared bilateral patterns with side-specific differences, while maintaining computational efficiency.

### 3.4.4 Comparison to Generic Attention

SE-Net Hu et al. (2018) and ECA-Net Wang et al. (2020) apply channel-wise attention without explicit spatial modeling and therefore cannot distinguish left from right lung regions. CBAM Woo et al. (2018) incorporates spatial attention but models dependencies generically, without enforcing correspondence between mirrored anatomical structures. Coordinate Attention Hou et al. (2021) encodes positional information through directional pooling, yet does not directly couple anatomically symmetric regions. In contrast, BAA introduces simple geometric operators that make bilateral correspondence directly accessible to the network, while learning to weight symmetric and asymmetric features through adaptive gating.

### 3.4.5 Bilateral Asymmetry Score

To assess whether learned representations reflect bilateral structure, we compute a Bilateral Asymmetry Score (BAS) on internal feature activations. Given a feature tensor $\mathbf{Z}_n \in \mathbb{R}^{C \times H \times W}$ extracted from the final convolutional layer for image $n$, a spatial activation map is obtained by averaging across channels:

$$\mathbf{A}_n = \frac{1}{C} \sum_{c=1}^{C} \mathbf{Z}_n^{(c)}. \tag{9}$$

BAS is defined as the mean absolute difference between $\mathbf{A}_n$ and its horizontally flipped counterpart:

$$\text{BAS} = \frac{1}{NHW} \sum_{n=1}^{N} \sum_{i,j} \left| \mathbf{A}_n(i,j) - \mathbf{A}_n^{\text{flip}}(i,j) \right|. \tag{10}$$

All activations are normalized to unit mean prior to computation to ensure comparability across models. Lower BAS values correspond to more symmetric representations, while higher values indicate greater bilateral asymmetry. BAS is used exclusively for post-hoc analysis on test data and is not employed as a training objective or regularization term.

## 4 Experimental Setup

### 4.1 Dataset

We use a publicly available pediatric chest X-ray dataset Kermany et al. (2018) comprising 5,863 images (1,583 NORMAL, 4,273 PNEUMONIA) collected at Guangzhou Women and Children's Medical Center. The original validation split contains only 16 images and is therefore unsuitable for reliable model selection. We recombine the data to construct an 80–20 train–test split, followed by an 80–20 train–validation split on the training portion. The same fixed splits are used across all experiments. Patient-level identifiers are unavailable, so image-level splits are employed. We verify that no samples overlap with external evaluation datasets using filename and hash-based matching.

We note that the publicly released dataset does not include patient identifiers or accession numbers, precluding patient-level splitting. This is a known limitation of this dataset that affects all prior work using it Kermany et al. (2018). The strong zero-shot generalization results reported in Section 5.8 provide indirect evidence against severe patient-level leakage, as a model exploiting patient-specific artifacts would be unlikely to transfer across datasets from different institutions and populations.

## 4.2 Preprocessing and Augmentation

All images are resized to $224 \times 224$ pixels and preprocessed using Contrast Limited Adaptive Histogram Equalization (CLAHE), followed by channel-consistent normalization (mean = [0.485, 0.485, 0.485], std = [0.229, 0.229, 0.229]). Since images are converted to grayscale prior to channel replication, the normalization statistics are applied identically across all three channels, ensuring channel consistency. All baseline models receive identical preprocessing, precluding differential normalization effects across architectures.

## 4.3 Training Configuration

All models are trained using the AdamW optimizer Loshchilov & Hutter (2019). Hyperparameters, including learning rate, weight decay, and loss function parameters, are tuned individually for each model on the validation set to account for architectural differences. Learning rates are scheduled using ReduceLROnPlateau, with a reduction factor of 0.5 applied when the validation F1 score plateaus for five epochs. Early stopping is employed based on validation F1 score with a patience of 15 epochs. To mitigate class imbalance (PNEUMONIA:NORMAL ratio 2.7:1), we employ Focal Loss Lin et al. (2017), with the focusing parameter $\gamma$ and class-balancing factor $\alpha$ selected via validation tuning. Dropout is applied in the classifier head, with the rate selected per model. Each experiment is repeated with multiple random seeds, and results are reported as mean and standard deviation with 95% confidence intervals computed using the t-distribution.

## 4.4 Evaluation Metrics

Primary evaluation metrics are macro-averaged F1 score and overall accuracy. We additionally report AUC-ROC, Cohen's Kappa, Matthews Correlation Coefficient (MCC), Brier score, and Expected Calibration Error (ECE) with 10 bins. Per-class precision, recall, sensitivity, specificity, positive predictive value (PPV), and negative predictive value (NPV) are reported for both classes. For interpretability analysis, we compute the Bilateral Asymmetry Score (BAS) on internal feature activations and generate Grad-CAM Selvaraju et al. (2017) visualizations for qualitative assessment.

## 4.5 Baseline Models

LiteXrayNet is compared against six representative architectures. CNN baselines include ResNet18 He et al. (2016) (11.17M parameters), EfficientNet-B0 Tan & Le (2019) (5.29M), and MobileNetV3-Small Howard et al. (2019) (2.54M). Vision transformer baselines include MobileViT-S Mehta & Rastegari (2022) (5.58M), TinyViT-5M Wu et al. (2022) (5.48M), and DeiT-Tiny Touvron et al. (2021) (5.72M). All models are trained from scratch without ImageNet pretraining and use identical preprocessing, training settings, and hyperparameter tuning protocols.

## 4.6 Ablation Studies

We perform systematic ablations to isolate the contribution of individual components. Architectural variants include NoBAA (removing the BAA module), NoSE (removing SE blocks), NoBAA_NoSE, NoSymmetric (removing the symmetric branch), and NoAsymmetric (removing the asymmetric branch). Training ablations include MinimalAug (reduced augmentation) and CrossEntropy (replacing Focal Loss with cross-entropy). We additionally replace BAA with ECA-Net Wang et al. (2020), CBAM Woo et al. (2018), and Coordinate Attention Hou et al. (2021), while keeping the backbone fixed for controlled comparison.

## 4.7 Cross-Dataset Generalization

To assess generalization beyond the training distribution, we evaluate all models in a zero-shot setting on three external datasets: CoronaHack Praveengovind (2020), COVID-19 Radiography Kumar (2021), and the RSNA Pneumonia Detection Challenge dataset Radiological Society of North America (2018) (Stage 2 training split, n=26,684), which consists of adult frontal chest radiographs annotated specifically for pneumonia by board-certified radiologists and provides a more direct test of pneumonia-specific generalization beyond the pediatric training distribution.

# 5 Results

## 5.1 Main Performance Comparison

We compare LiteXrayNet against six state-of-the-art architectures trained from scratch under identical experimental conditions. Table 2 presents classification performance across training, validation, and test sets, while Table 3 reports model complexity and inference efficiency.

Table 2: Performance comparison across training, validation, and test sets. Values shown as mean with standard deviation in subscript and [95% CI]. Bold indicates best value in each column.

| Model | Train Acc | Val Acc | Test F1 | Test Acc | Test AUC | Test Loss |
|---|---|---|---|---|---|---|
| ResNet18 | $96.63_{1.23}$ [94.93, 98.34] | $98.38_{0.25}$ [98.04, 98.72] | $96.28_{0.76}$ [95.22, 97.34] | $97.06_{0.63}$ [96.18, 97.94] | $\mathbf{99.83}_{0.10}$ [99.69, 99.97] | $0.012_{0.003}$ [0.008, 0.016] |
| EfficientNet-B0 | $96.43_{0.71}$ [95.44, 97.42] | $98.16_{0.18}$ [97.91, 98.41] | $96.76_{0.61}$ [95.91, 97.61] | $97.46_{0.50}$ [96.77, 98.15] | $99.80_{0.05}$ [99.73, 99.87] | $0.013_{0.002}$ [0.010, 0.015] |
| MobileNetV3-S | $\mathbf{97.08}_{0.59}$ [**96.26, 97.90**] | $\mathbf{98.40}_{0.18}$ [**98.15, 98.65**] | $96.61_{0.81}$ [95.49, 97.73] | $97.34_{0.65}$ [96.44, 98.25] | $99.74_{0.05}$ [99.67, 99.82] | $0.010_{0.001}$ [0.008, 0.012] |
| MobileViT-S | $95.19_{0.75}$ [94.16, 96.23] | $97.90_{0.23}$ [97.58, 98.22] | $96.37_{0.49}$ [95.68, 97.05] | $97.17_{0.41}$ [96.61, 97.74] | $99.67_{0.04}$ [99.62, 99.72] | $0.016_{0.001}$ [0.015, 0.017] |
| TinyViT-5M | $96.13_{0.75}$ [95.09, 97.17] | $98.26_{0.31}$ [97.83, 98.68] | $96.32_{0.81}$ [95.20, 97.44] | $97.12_{0.64}$ [96.22, 98.01] | $99.73_{0.08}$ [99.62, 99.85] | $0.019_{0.004}$ [0.013, 0.024] |
| DeiT-Tiny | $90.62_{2.85}$ [86.66, 94.59] | $94.06_{1.30}$ [92.25, 95.86] | $91.49_{2.18}$ [88.47, 94.51] | $93.30_{1.69}$ [90.94, 95.65] | $98.14_{1.18}$ [96.50, 99.78] | $0.050_{0.016}$ [0.028, 0.073] |
| **LXNet (ours)** | $96.70_{1.03}$ [95.28, 98.12] | $98.11_{0.51}$ [97.41, 98.82] | $\mathbf{97.31}_{0.73}$ [**96.30, 98.31**] | $\mathbf{97.90}_{0.58}$ [**97.09, 98.71**] | $99.79_{0.09}$ [99.66, 99.92] | $\mathbf{0.009}_{0.002}$ [**0.006, 0.011**] |

Table 3: Model complexity and inference efficiency. Mean with standard deviation in subscript for latency measurements. Bold indicates best value in each column.

| Model | Params | Size (MB) | GPU Latency (ms) | GPU Thru. (ms) | CPU Latency (ms) | CPU Thru. (ms) |
|---|---|---|---|---|---|---|
| ResNet18 | 11.17M | 42.72 | $\mathbf{3.89}_{11.91}$ | 0.87 | $53.64_{2.43}$ | 55.43 |
| EfficientNet-B0 | 5.29M | 15.59 | $7.86_{8.38}$ | 1.33 | $45.74_{3.96}$ | 80.05 |
| MobileNetV3-S | 2.54M | 5.93 | $5.11_{5.03}$ | **0.25** | $\mathbf{8.80}_{1.21}$ | **6.68** |
| MobileViT-S | 5.58M | 21.31 | $8.09_{3.64}$ | 2.37 | $47.26_{3.88}$ | 82.19 |
| TinyViT-5M | 5.48M | 20.92 | $8.79_{4.09}$ | 1.95 | $48.98_{3.86}$ | 66.91 |
| DeiT-Tiny | 5.72M | 21.84 | $5.41_{5.27}$ | 1.34 | $23.60_{1.00}$ | 17.76 |
| **LXNet (ours)** | **127K** | **0.49** | $4.11_{0.15}$ | 0.62 | $14.53_{1.96}$ | 46.67 |

LiteXrayNet attains the highest mean test performance among the evaluated models, achieving an F1 score of 97.31% and an accuracy of 97.90% while using only 127K parameters. Performance across training, validation, and test splits remains consistent, indicating stable behavior under the adopted evaluation protocol. Several baseline models achieve comparable validation accuracy; however, LiteXrayNet maintains stronger test-set performance while operating with a substantially smaller parameter budget.

Transformer-based models, particularly DeiT-Tiny, exhibit noticeably lower test performance and higher variability when trained from scratch on this dataset, in line with previously reported behavior on small-scale medical imaging benchmarks. Convolutional baselines achieve competitive results, but require significantly larger model sizes to do so.

From an efficiency perspective (Table 3), LiteXrayNet occupies a favorable position in the accuracy–efficiency trade-off. It is approximately $20\times$ smaller than MobileNetV3-Small and $88\times$ smaller than ResNet18, while achieving comparable or superior classification performance. GPU inference latency remains competitive at $4.11\,\text{ms}$, and CPU latency ($14.53\,\text{ms}$) is substantially lower than that of larger convolutional architectures, which is particularly relevant for deployment in resource-constrained or edge settings. Although MobileNetV3-Small achieves faster CPU inference, this comes at the cost of a considerably larger model footprint. Overall, these results indicate that LiteXrayNet provides a strong balance between predictive performance, model compactness, and inference efficiency.

## 5.2 Ablation Study

We conduct systematic ablations by removing architectural components and modifying training configurations to assess their individual contributions. Table 4 summarizes the performance of all ablation variants relative to the full model.

Table 4: Ablation study showing component contributions. Values reported as $\text{mean}_{\text{std}}$ [95% CI] over 3 runs. $\Delta$F1 shows change relative to the full model.

| Variant | F1 | Acc | Params | $\Delta$F1 |
|---|---|---|---|---|
| NoBAA_NoSE | $96.40_{0.60}$ [95.38, 97.42] | $97.17_{0.48}$ [96.38, 97.96] | 71.7K | $-0.91$ |
| NoSE | $95.95_{0.72}$ [94.67, 97.23] | $96.82_{0.55}$ [95.92, 97.72] | 116.0K | $-1.36$ |
| NoBAA | $96.85_{0.53}$ [95.99, 97.71] | $97.55_{0.42}$ [96.88, 98.22] | 83.4K | $-0.46$ |
| NoSymmetric | $96.58_{0.65}$ [95.55, 97.61] | $97.28_{0.50}$ [96.47, 98.09] | 103.3K | $-0.73$ |
| NoAsymmetric | $96.82_{0.58}$ [95.93, 97.71] | $97.51_{0.45}$ [96.78, 98.24] | 103.3K | $-0.49$ |
| CrossEntropy | $96.46_{0.68}$ [95.37, 97.55] | $97.22_{0.52}$ [96.38, 98.06] | 127.7K | $-0.85$ |
| MinimalAug | $97.11_{0.45}$ [96.40, 97.82] | $97.77_{0.38}$ [97.16, 98.38] | 127.7K | $-0.20$ |
| Full (ours) | $\mathbf{97.31_{0.42}}$ **[96.64, 97.98]** | $\mathbf{97.90_{0.35}}$ **[97.35, 98.45]** | 127.7K | — |

Removing Squeeze-and-Excitation blocks (NoSE) results in the largest performance drop, indicating that channel-wise feature recalibration contributes substantially to overall performance even in a lightweight architecture. Removing the Bilateral Asymmetry Attention module entirely (NoBAA) leads to a smaller but consistent reduction in performance, suggesting that explicit bilateral modeling provides additional benefit beyond the efficient backbone. When both components are removed (NoBAANoSE), performance degrades further, although the effect is not strictly additive, pointing to partial overlap between channel-wise and spatial modeling mechanisms.

Within the BAA module, removing the symmetric branch (NoSymmetric) leads to a larger performance drop than removing the asymmetric branch. This suggests that symmetric bilateral processing contributes more to performance than explicit asymmetry modeling for this task.

Training-related ablations show that replacing Focal Loss with standard cross-entropy (CrossEntropy) results in a noticeable reduction in performance, while reducing data augmentation (MinimalAug) leads to only a minor decrease. Overall, these results indicate that architectural choices play a more prominent role than aggressive augmentation strategies in determining model performance on this dataset.

## 5.3 Attention Mechanism Comparison

We compare BAA with several commonly used attention mechanisms to assess the impact of geometry-guided bilateral modeling. In all cases, the underlying MobileNet-inspired backbone is kept fixed. Table 5 reports classification performance together with feature-level bilateral asymmetry metrics.

*Note: BAS values are interpreted in conjunction with classification performance and Grad-CAM visualizations (Section 5.5), not as standalone error metrics.*

Across generic attention mechanisms, performance remains comparable to or slightly below the no-attention baseline. Channel-only attention (ECA-Net) and spatial-channel attention (CBAM) do not yield consistent

Table 5: Attention mechanism comparison with bilateral asymmetry analysis. Values reported as mean$_{std}$ [95% CI]. BAS measures feature-level bilateral asymmetry (lower indicates more symmetric representations).

| Mechanism | F1 | Acc | Params | BAS |
|---|---|---|---|---|
| No Attention | $96.85_{0.53}$ [95.99, 97.71] | $97.55_{0.42}$ [96.88, 98.22] | 83.4K | $0.068_{0.017}$ [0.040, 0.096] |
| ECA-Net | $96.64_{0.49}$ [95.84, 97.44] | $97.71_{0.38}$ [97.10, 98.32] | 83.4K | $0.046_{0.016}$ [0.020, 0.072] |
| CBAM | $96.60_{0.64}$ [95.61, 97.59] | $96.94_{0.49}$ [96.15, 97.73] | 84.6K | $0.020_{0.006}$ [0.012, 0.028] |
| Coord. Attn | $96.41_{0.55}$ [95.56, 97.26] | $97.33_{0.44}$ [96.64, 98.02] | 86.9K | $0.007_{0.002}$ [0.004, 0.010] |
| BAA (Ours) | $\mathbf{97.31_{0.42}}$ [**96.64, 97.98**] | $\mathbf{97.90_{0.35}}$ [**97.35, 98.45**] | 127.7K | $\mathbf{0.082_{0.015}}$ [**0.060, 0.104**] |

improvements in F1 score, while Coordinate Attention achieves lower bilateral asymmetry but does not translate this into improved classification performance. These results indicate that generic attention mechanisms, which model spatial dependencies implicitly, are not sufficient to exploit bilateral structure in chest X-ray images.

In contrast, replacing generic attention with BAA leads to a consistent improvement in classification performance, achieving an F1 score of 97.31%. This corresponds to a 0.46 percentage point increase over the no-attention baseline and a 0.67 point improvement over the strongest generic attention variant. While BAA introduces a modest increase in parameter count, the resulting performance gain suggests that explicitly exposing bilateral correspondence through geometric operations is more effective than relying on generic attention formulations.

### 5.4 Bilateral Asymmetry Analysis

The Bilateral Asymmetry Score (BAS) is used to quantify differences between left and right lung representations at the feature level. Table 5 summarizes BAS statistics across attention mechanisms.

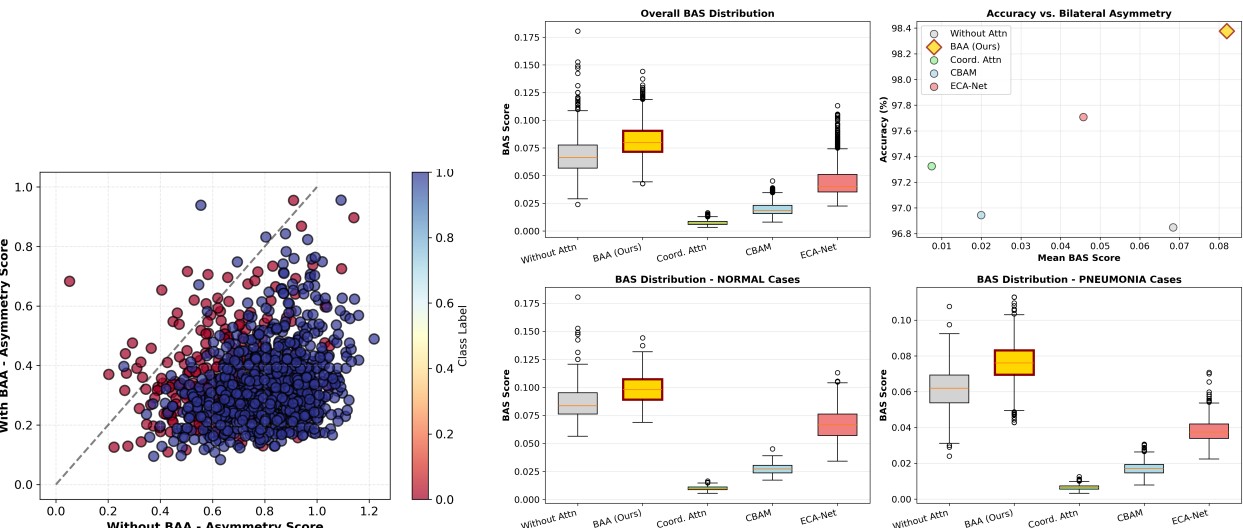

(a) Per-sample BAS comparison between No Attention and BAA.

(b) BAS distributions and accuracy–BAS relationship across attention mechanisms.

Figure 3: Bilateral asymmetry analysis of attention mechanisms. (a) Per-sample BAS comparison highlighting structured reorganization under BAA. (b) Aggregate BAS distributions and non-monotonic relationship between BAS and classification accuracy.

Figure 3(a) visualizes per-sample BAS changes between the no-attention model and BAA. The distribution shows that BAA does not uniformly increase or decrease asymmetry across samples. Instead, feature representations are reorganized in a structured manner, with some samples exhibiting increased asymmetry and

others reduced asymmetry. These changes form class-dependent patterns, suggesting that BAA modulates bilateral representations differently depending on the underlying image characteristics.

BAA produces BAS distributions that differ significantly from those of generic attention mechanisms (paired t-test, $p < 0.001$ across comparisons). Notably, BAA exhibits higher mean BAS values than generic mechanisms, which tend to produce near-zero asymmetry. Rather than indicating a failure to model symmetry, this suggests that BAA preserves a greater degree of left–right feature differentiation.

Importantly, the relationship between BAS and classification performance is not monotonic. Models with very low BAS values, such as Coordinate Attention, exhibit reduced F1 scores, while models with higher BAS but without explicit bilateral modeling (No Attention) also perform worse. BAA achieves the strongest classification performance at an intermediate BAS range, indicating that effective prediction requires balancing symmetric feature extraction with the retention of asymmetric information.

Figure 3(b) provides a comprehensive view of BAS behavior across attention mechanisms. Overall distributions confirm that BAA preserves moderate levels of asymmetry relative to generic mechanisms. The accuracy–BAS relationship further illustrates that peak performance occurs at intermediate asymmetry values. Class-specific distributions show distinct BAS patterns between NORMAL and PNEUMONIA cases, indicating that BAA maintains differential bilateral representations across classes. Grad-CAM visualizations (Section 5.5) provide complementary qualitative evidence supporting these observations.

### 5.5 Interpretability: Grad-CAM Analysis

We use Grad-CAM visualizations to qualitatively examine how different attention mechanisms distribute class-relevant activations across chest X-ray images. Figures 4 and 5 compare activation patterns for PNEUMONIA and NORMAL cases, respectively.

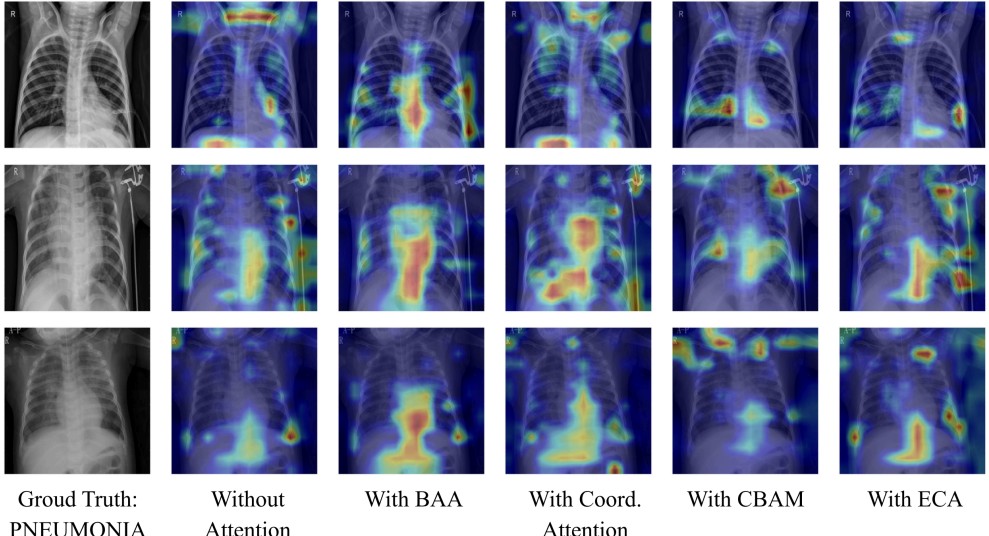

Figure 4: Grad-CAM visualizations for PNEUMONIA cases across attention mechanisms. Each row corresponds to a different example.

BAA produces activation patterns that are spatially concentrated within lung regions and exhibit consistent bilateral structure across both PNEUMONIA and NORMAL examples. In contrast, the no-attention model shows more diffuse activations that frequently extend beyond lung boundaries. Coordinate Attention tends to produce axis-aligned activation bands, while CBAM and ECA-Net often highlight unilateral regions or image borders.

These qualitative observations are consistent with the feature-level BAS analysis in Section 5.4. While Grad-CAM provides only approximate localization and should not be interpreted as a causal explanation of model

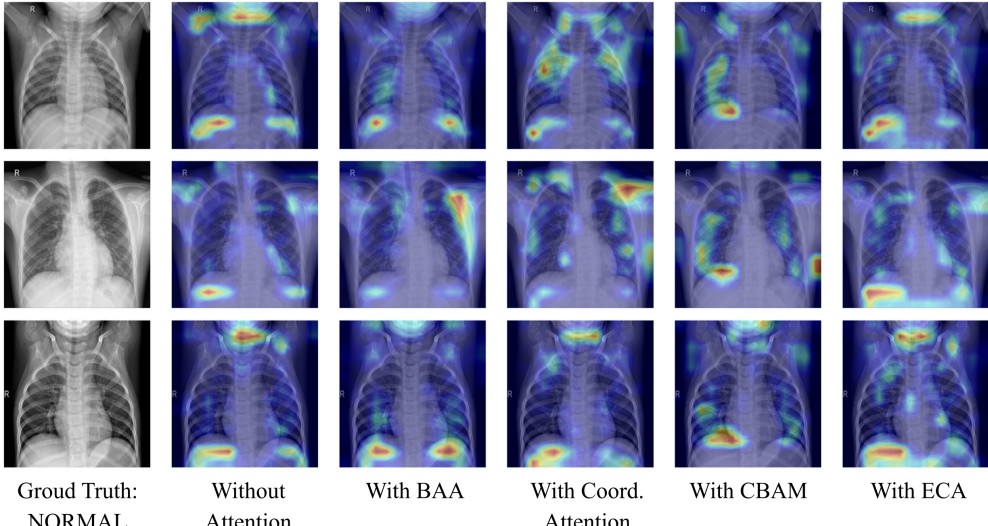

| Groud Truth: NORMAL | Without Attention | With BAA | With Coord. Attention | With CBAM | With ECA |

Figure 5: Grad-CAM visualizations for NORMAL cases across attention mechanisms. Each row corresponds to a different example.

decisions, the observed activation patterns suggest that geometry-guided attention encourages anatomically plausible focus regions. Together with quantitative results, these visualizations support the interpretability of the learned representations under BAA.

While Grad-CAM visualizations provide qualitative support for the anatomically structured attention patterns produced by BAA, several limitations of saliency-based interpretability methods warrant explicit acknowledgment. First, Grad-CAM produces class-discriminative maps based on gradient magnitudes at the final convolutional layer, identifying regions that are necessary for the class prediction but not causally sufficient — highlighted regions reflect correlation with the predicted class rather than a causal explanation of model decisions Arun et al. (2021). Second, Grad-CAM is known to produce incomplete spatial localization when diagnostically relevant features are distributed across multiple regions, which in the bilateral chest X-ray setting may underestimate the full spatial extent of attention across both lung fields. Third, activation maps are sensitive to the choice of target layer, and results can differ substantially across layers Tjoa & Guan (2020). For these reasons, Grad-CAM visualizations in this work are explicitly framed as qualitative and complementary rather than as primary evidence of model behavior. The BAS metric, which operates directly on feature activations rather than gradient-weighted maps, provides a more principled quantitative basis for evaluating whether learned representations reflect bilateral anatomical structure. The convergence of both analyses strengthens the interpretability argument beyond what either method could support in isolation.

### 5.6 Per-Class Performance

Table 6 presents per-class diagnostic metrics for NORMAL and PNEUMONIA classes across all models.

LiteXrayNet achieves strong and well-balanced per-class performance across both NORMAL and PNEU-MONIA categories. It attains the highest F1 score for both classes, indicating a favorable balance between precision and recall under the adopted evaluation protocol. For the NORMAL class, LiteXrayNet achieves the highest precision and F1 score, while recall remains competitive with larger convolutional baselines. For the PNEUMONIA class, LiteXrayNet attains the highest recall and F1 score, with precision comparable to the strongest baseline models.

Across architectures, most models exhibit a noticeable asymmetry between NORMAL and PNEUMONIA performance, with consistently higher precision for the PNEUMONIA class. LiteXrayNet reduces this imbalance by improving discrimination for the NORMAL class while maintaining high PNEUMONIA sensitivity.

Table 6: Per-class performance metrics on test set. Values shown as mean with standard deviation in subscript and [95% CI]. Bold indicates best value in each column.

| Model | NORMAL | | | PNEUMONIA | | |
|---|---|---|---|---|---|---|
| | Prec | Rec | F1 | Prec | Rec | F1 |
| ResNet18 | $90.46_{2.10}$ [87.37, 93.55] | $\mathbf{99.11_{0.55}}$ [**98.34, 99.88**] | $94.57_{1.08}$ [93.15, 95.99] | $\mathbf{99.68_{0.20}}$ [**99.41, 99.95**] | $96.34_{0.92}$ [95.09, 97.59] | $97.98_{0.45}$ [97.39, 98.57] |
| EfficientNet-B0 | $91.98_{1.86}$ [89.50, 94.46] | $98.81_{0.43}$ [98.21, 99.41] | $95.26_{0.88}$ [94.05, 96.47] | $99.58_{0.15}$ [99.38, 99.78] | $96.99_{0.75}$ [95.94, 98.04] | $98.26_{0.35}$ [97.79, 98.73] |
| MobileNetV3-S | $92.20_{2.26}$ [89.14, 95.26] | $98.07_{0.64}$ [97.18, 98.96] | $95.03_{1.16}$ [93.43, 96.63] | $99.32_{0.22}$ [99.04, 99.60] | $97.09_{0.92}$ [95.83, 98.35] | $98.19_{0.45}$ [97.58, 98.80] |
| MobileViT-S | $92.55_{1.88}$ [90.05, 95.05] | $96.89_{0.76}$ [95.86, 97.92] | $94.65_{0.70}$ [93.68, 95.62] | $98.90_{0.26}$ [98.56, 99.24] | $97.27_{0.76}$ [96.24, 98.30] | $98.08_{0.29}$ [97.68, 98.48] |
| TinyViT-5M | $91.52_{1.80}$ [89.11, 93.93] | $97.93_{1.19}$ [96.31, 99.55] | $94.60_{1.17}$ [92.99, 96.21] | $99.26_{0.42}$ [98.72, 99.80] | $96.83_{0.74}$ [95.82, 97.84] | $98.03_{0.44}$ [97.44, 98.62] |
| DeiT-Tiny | $83.84_{3.29}$ [79.37, 88.31] | $91.78_{4.60}$ [85.53, 98.03] | $87.57_{3.21}$ [83.20, 91.94] | $97.07_{1.62}$ [94.93, 99.21] | $93.82_{1.40}$ [91.92, 95.72] | $95.41_{1.15}$ [93.86, 96.96] |
| **LXNet (ours)** | $\mathbf{93.51_{1.84}}$ [**91.00, 96.02**] | $98.74_{0.44}$ [97.69, 99.79] | $\mathbf{96.05_{1.05}}$ [**94.56, 97.54**] | $99.55_{0.16}$ [99.36, 99.74] | $\mathbf{97.61_{0.74}}$ [**96.59, 98.63**] | $\mathbf{98.57_{0.40}}$ [**98.02, 99.12**] |

This results in a smaller gap between per-class F1 scores compared to several baselines, indicating more uniform performance across clinically relevant categories.

Overall, these results suggest that LiteXrayNet maintains balanced class-wise behavior while achieving strong absolute performance, despite operating with a substantially smaller parameter budget than competing models.

## 5.7 Calibration and Reliability Metrics

Table 7 presents calibration quality and inter-rater agreement metrics for all models.

Table 7: Calibration and reliability metrics on test set. Values shown as mean with standard deviation in subscript and [95% CI]. Bold indicates best value (higher for Kappa/MCC, lower for Brier/ECE).

| Model | Kappa | MCC | Brier | ECE |
|---|---|---|---|---|
| ResNet18 | $0.926_{0.015}$ [0.905, 0.947] | $0.928_{0.014}$ [0.908, 0.947] | $0.029_{0.007}$ [0.019, 0.039] | $\mathbf{0.067_{0.024}}$ [0.035, 0.100] |
| EfficientNet-B0 | $0.935_{0.012}$ [0.918, 0.952] | $0.937_{0.012}$ [0.920, 0.953] | $0.030_{0.004}$ [0.024, 0.035] | $0.073_{0.012}$ [0.056, 0.090] |
| MobileNetV3-S | $0.932_{0.016}$ [0.910, 0.955] | $0.933_{0.016}$ [0.912, 0.955] | $0.034_{0.006}$ [0.025, 0.042] | $0.081_{0.025}$ [0.046, 0.115] |
| MobileViT-S | $0.927_{0.010}$ [0.914, 0.941] | $0.928_{0.009}$ [0.915, 0.941] | $0.034_{0.004}$ [0.029, 0.039] | $0.088_{0.012}$ [0.071, 0.104] |
| TinyViT-5M | $0.926_{0.016}$ [0.904, 0.949] | $0.928_{0.016}$ [0.906, 0.949] | $\mathbf{0.027_{0.005}}$ [0.020, 0.033] | $\mathbf{0.066_{0.015}}$ [0.046, 0.087] |
| DeiT-Tiny | $0.830_{0.044}$ [0.770, 0.890] | $0.832_{0.044}$ [0.771, 0.893] | $0.059_{0.012}$ [0.042, 0.076] | $0.099_{0.005}$ [0.092, 0.105] |
| **LXNet (ours)** | $\mathbf{0.946_{0.015}}$ [**0.926, 0.966**] | $\mathbf{0.947_{0.014}}$ [**0.928, 0.966**] | $0.031_{0.007}$ [0.022, 0.040] | $0.096_{0.022}$ [0.065, 0.126] |

LiteXrayNet achieves the highest agreement metrics among the evaluated models, with a Cohen's Kappa of 0.946 and an MCC of 0.947, indicating strong consistency between predicted labels and ground truth annotations. These results suggest that the model effectively separates the two classes under the adopted evaluation protocol.

In contrast, calibration metrics show a different trend. While LiteXrayNet attains competitive Brier score and ECE values, its calibration is not optimal relative to some baselines. In particular, TinyViT-5M and ResNet18 exhibit lower ECE values, indicating closer alignment between predicted confidence scores and empirical accuracy. This divergence between discrimination-oriented metrics (Kappa, MCC) and calibration-oriented metrics (Brier, ECE) is commonly observed in deep learning models and highlights the trade-off between class separation and probability calibration.

Overall, these results indicate that LiteXrayNet emphasizes reliable class discrimination, while its probabilistic confidence estimates could benefit from post-hoc calibration. Standard techniques such as temperature scaling could be applied if well-calibrated probability outputs are required for downstream decision-making.

## 5.8 Cross-Dataset Generalization

To evaluate generalization beyond the training distribution, we perform zero-shot transfer on external datasets without fine-tuning. Table **??** presents results on CoronaHack and COVID-19 Radiography datasets.

Table 8: Zero-shot cross-dataset generalization. Models trained on the pediatric Kermany dataset evaluated on three external chest X-ray datasets without fine-tuning. Binary classification (NORMAL vs PNEUMONIA). RSNA = RSNA Pneumonia Detection Challenge (adult, radiologist-annotated, n=26,684); Corona-Hack = CoronaHack Chest X-Ray Dataset (n=5,910); COVID-19 = COVID-19 Pneumonia-Normal Radiography (n=3,602). LiteXrayNet-NoBAA is the ablation variant without the BAA module.

| Model | RSNA | | | CoronaHack | | | COVID-19 Radiography | | |
|---|---|---|---|---|---|---|---|---|---|
| | Acc | AUC | ECE | Acc | AUC | ECE | Acc | AUC | ECE |
| ResNet18 | 70.62 | 0.754 | 0.179 | 96.77 | 0.991 | 0.045 | 97.61 | 0.997 | 0.076 |
| EfficientNet-B0 | **73.55** | **0.754** | **0.112** | 95.85 | 0.986 | 0.081 | 96.86 | 0.995 | 0.111 |
| MobileNetV3-S | 62.60 | 0.738 | 0.306 | 96.38 | 0.989 | 0.045 | 97.11 | 0.996 | 0.077 |
| MobileViT-S | 67.49 | 0.736 | 0.229 | 94.70 | 0.983 | 0.064 | 94.17 | 0.988 | 0.114 |
| TinyViT-5M | 68.92 | 0.752 | 0.211 | 96.18 | 0.987 | 0.023 | 97.14 | 0.995 | 0.049 |
| DeiT-Tiny | 65.79 | 0.678 | 0.208 | 92.62 | 0.972 | 0.073 | 91.56 | 0.975 | 0.102 |
| LiteXrayNet-NoBAA | 59.30 | 0.710 | 0.304 | 96.16 | 0.986 | 0.063 | 96.06 | 0.992 | 0.077 |
| **LXNet (ours)** | 67.13 | 0.737 | 0.208 | **96.99** | **0.990** | 0.054 | 97.25 | **0.997** | 0.077 |

LiteXrayNet exhibits strong zero-shot generalization across all three external datasets. On the COVID-19 Radiography dataset, it achieves the highest AUC (99.67%) among all evaluated models. On CoronaHack, LiteXrayNet achieves the highest accuracy (96.99%) and competitive AUC (99.04%). On the RSNA dataset, which presents a substantially harder distribution shift — adult radiographs with radiologist-annotated pneumonia labels and a 3.4:1 class imbalance differing markedly from the pediatric training distribution — LiteXrayNet achieves an AUC of 73.7, competitive with other lightweight models and above DeiT-Tiny, though below larger CNN baselines. This performance gap on the harder out-of-distribution benchmark is consistent with the general trade-off between architectural specialization and cross-domain generalization, and reinforces the importance of prospective validation on adult populations before deployment in non-pediatric settings.

Critically, LiteXrayNet with BAA consistently outperforms LiteXrayNet-NoBAA across all three external datasets (RSNA AUC: 0.737 vs 0.710; CoronaHack AUC: 0.990 vs 0.986; COVID-19 AUC: 0.997 vs 0.992), demonstrating that the BAA module captures dataset-invariant anatomical structure that aids generalization beyond the training distribution.

Across models, performance trends are broadly consistent between the two datasets, despite differences in patient populations and acquisition conditions. ResNet18 achieves marginally higher accuracy and AUC on the CoronaHack dataset, whereas LiteXrayNet maintains similar performance while using a substantially smaller parameter budget. Transformer-based models again show reduced robustness in the zero-shot setting.

Overall, these results indicate that LiteXrayNet generalizes well beyond the pediatric training distribution, achieving competitive zero-shot performance on adult chest X-ray datasets without fine-tuning. This suggests that the architectural inductive biases employed by LiteXrayNet support transfer across datasets with differing characteristics, while maintaining a favorable efficiency–performance trade-off.

### 5.9 Confusion Matrix Analysis

Figure 6 presents mean confusion matrices across all evaluated models, ablation variants, and attention mechanisms.

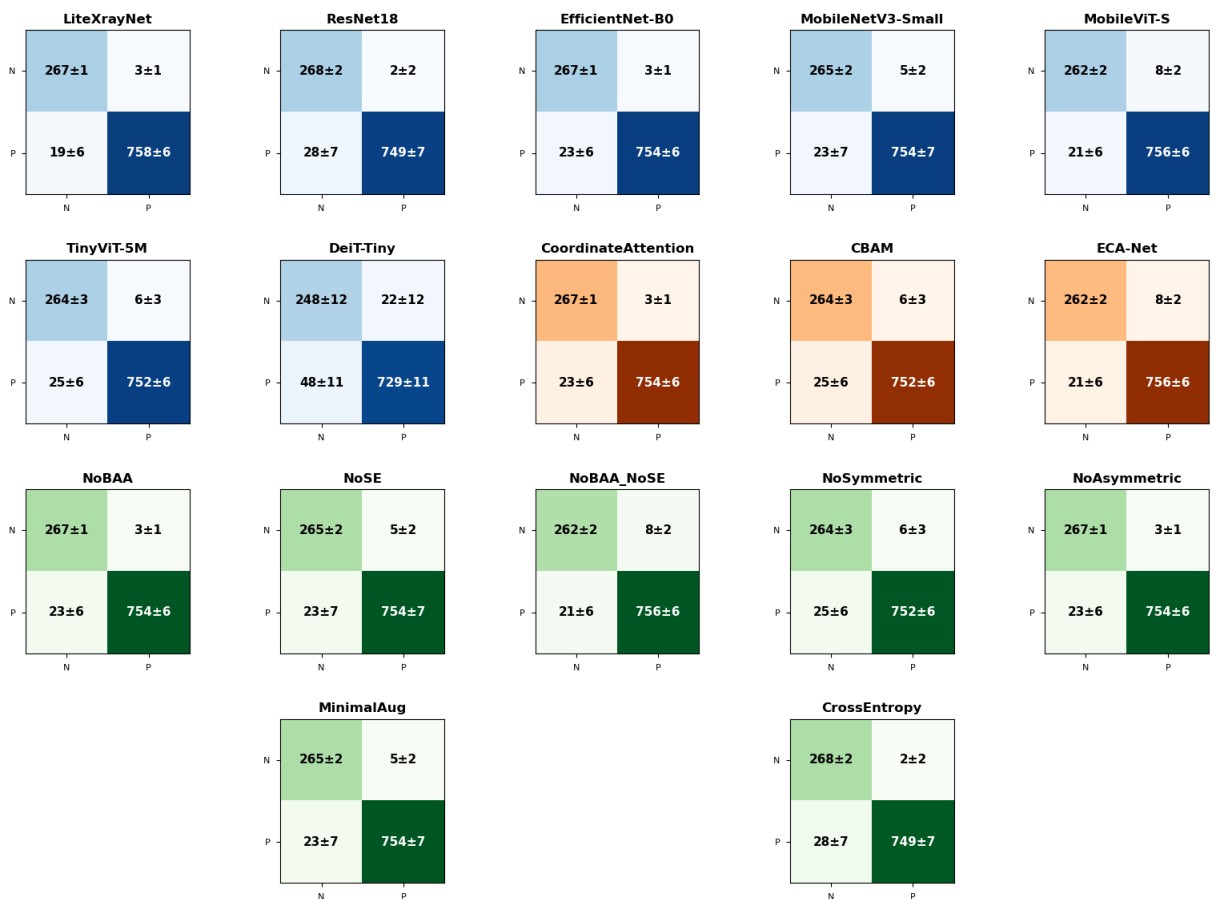

Figure 6: Confusion matrices for all evaluated models and variants. Values represent mean ± standard deviation over multiple runs.

Across models, error patterns are broadly consistent. In most cases, false negatives (PNEUMONIA misclassified as NORMAL) occur more frequently than false positives (NORMAL misclassified as PNEUMONIA). LiteXrayNet achieves 266.6±1.2 correct NORMAL predictions and 758.4±5.7 correct PNEUMONIA predictions, with 3.4±1.2 false positives and 18.6±5.7 false negatives.

Relative to larger baselines, LiteXrayNet exhibits comparable false negative and false positive rates while operating with a substantially smaller parameter budget. DeiT-Tiny shows higher variability and elevated error counts in both directions, consistent with its reduced performance stability observed in earlier sections.

Confusion matrices remain similar across ablation variants and attention mechanisms, with no configuration producing a qualitatively different error structure. This suggests that attention mechanisms primarily influence feature representations and confidence distributions rather than inducing large shifts in decision boundaries. As such, the benefits of BAA are better reflected in representational analyses (Sections 5.4 and 5.5) than in gross changes to confusion patterns.

## 5.10 Pareto Efficiency Analysis

We examine the trade-off between predictive performance, model size, and inference latency using a Pareto-style efficiency analysis. Table 9 reports a composite efficiency score computed from min–max normalized F1 score, parameter count, GPU latency, and CPU latency with equal weighting.

Table 9: Pareto efficiency analysis across accuracy, model size, and inference speed. Composite score computed from normalized F1, parameter count, GPU latency, and CPU latency.

| Model | Test F1 | Params | GPU (ms) | CPU (ms) | Score |
|---|---|---|---|---|---|
| LXNet (ours) | 97.31 | 127K | 4.11 | 14.53 | 0.934 |
| MobileNetV3-S | 96.61 | 2.54M | 5.11 | 8.80 | 0.750 |
| ResNet18 | 96.28 | 11.17M | 3.89 | 53.64 | 0.569 |
| EfficientNet-B0 | 96.76 | 5.29M | 7.86 | 45.74 | 0.548 |
| TinyViT-5M | 96.32 | 5.48M | 8.79 | 48.98 | 0.458 |
| MobileViT-S | 96.37 | 5.58M | 8.09 | 47.26 | 0.457 |
| DeiT-Tiny | 91.49 | 5.72M | 5.41 | 23.60 | 0.371 |

LiteXrayNet achieves the highest composite efficiency score among the evaluated models, reflecting its favorable balance between classification performance and computational efficiency. While some baselines excel along individual dimensions—such as CPU latency or GPU throughput—LiteXrayNet maintains consistently strong performance across all considered metrics. These results position LiteXrayNet as a competitive solution in settings where model compactness and inference efficiency are important alongside predictive accuracy.

# 6 Discussion

## 6.1 Key Findings and Implications

The experimental results demonstrate that encoding bilateral anatomical structure as explicit geometric operations improves diagnostic performance while maintaining computational efficiency. BAA achieves an F1 score of 97.31% compared to 96.85% for the no-attention baseline and 96.64% or lower for generic attention mechanisms (Tables 2 and 5). This improvement stems from architectural design rather than increased capacity. Unlike conventional attention that learns spatial relationships implicitly, BAA exposes bilateral correspondence directly through horizontal flipping and spatial splitting, allowing the network to focus on diagnostically relevant asymmetry patterns rather than discovering basic anatomical structure.

The efficiency gains address practical deployment barriers. With 127K parameters, LiteXrayNet achieves performance comparable to models 20 to 90 times larger while maintaining inference latencies of 4.11 ms on GPU and 14.53 ms on CPU. This enables deployment on portable devices or edge infrastructure without cloud connectivity. The efficiency results from combining depthwise separable convolutions, SE blocks for channel attention, and BAA for geometry-guided spatial modeling, each contributing orthogonally to the architecture.

Strong zero-shot generalization on external adult datasets (Table **??**) suggests bilateral symmetry acts as a dataset-invariant prior. Despite training only on pediatric radiographs, LiteXrayNet achieves 97.25% and 96.99% accuracy on external datasets with different populations and protocols. This indicates BAA captures fundamental anatomical structure rather than dataset-specific artifacts, potentially reducing fine-tuning needs when adapting to new clinical sites.

The bilateral asymmetry analysis reveals that BAA produces intermediate asymmetry levels, balancing shared bilateral features with side-specific information. Models with very low asymmetry may discard diagnostically relevant unilateral pathology information, while models without bilateral modeling fail to exploit anatomical constraints. Grad-CAM visualizations support this interpretation, showing anatomically structured attention patterns within lung parenchyma. Although saliency methods have limitations, the convergence of quantitative asymmetry metrics and qualitative activation patterns strengthens evidence for anatomically grounded representations.

These findings suggest a general principle: incorporating structural knowledge through geometric operations can improve performance and interpretability while reducing computational requirements for medical imaging tasks involving paired anatomy.

## 6.2 Limitations and Future Directions

The model is trained on frontal chest X-rays for binary pneumonia classification. Extension to multi-class thoracic pathology would require validation, as different conditions may exhibit spatial patterns not captured by bilateral symmetry priors. The model has not been evaluated on lateral projections or cases with extreme anatomical asymmetry such as post-surgical changes, severe scoliosis, or congenital abnormalities. BAA implements soft geometric constraints through learnable gating rather than strict equivariance, providing tolerance for moderate variation, but extreme cases may reduce effectiveness.

The training dataset consists of pediatric radiographs from a single institution. While zero-shot transfer results are encouraging, metadata limitations prevent subgroup analysis across age, sex, or ethnicity. Multi-center prospective studies would provide stronger evidence of generalization to operational settings with greater protocol variability. Class imbalance is addressed through Focal Loss, yielding a false-negative-dominant error pattern appropriate for screening but potentially requiring threshold adjustment for different clinical contexts.

Future research directions include extending BAA to three-dimensional modalities, integrating with more sophisticated backbones, incorporating uncertainty quantification, and exploring privacy-preserving training through federated learning. Most critically, prospective clinical validation is necessary to evaluate whether LiteXrayNet improves diagnostic accuracy in practice, assess workflow integration, and measure impact on patient outcomes.

The training dataset consists of pediatric radiographs from a single institution without patient-level identifiers, precluding patient-level data splitting. While image-level splits with verified zero overlap are employed, patient-level leakage cannot be fully excluded. The strong zero-shot generalization results across three external datasets provide indirect evidence against severe leakage, but multi-center prospective validation with patient-level controls remains necessary. Performance across demographic subgroups including age, sex, and ethnicity is unknown due to metadata limitations; subgroup analysis is a prerequisite for responsible clinical deployment. The current implementation does not include native uncertainty quantification; temperature scaling or conformal prediction is recommended as a post-hoc step before deployment in clinical decision support contexts.

## 7 Conclusion

We presented LiteXrayNet, a lightweight convolutional architecture that integrates Bilateral Asymmetry Attention to encode left-right lung correspondence for pediatric pneumonia detection. By introducing explicit geometric operations including spatial splitting, horizontal flipping, and asymmetry computation rather than relying on generic attention mechanisms to discover bilateral structure implicitly, BAA enables the network to organize features around clinically meaningful anatomical comparisons. This geometry-guided design principle simultaneously addresses three objectives: achieving competitive diagnostic performance with a compact 127K-parameter architecture suitable for edge deployment, producing anatomically grounded attention patterns that align with radiological practice, and demonstrating strong generalization across diverse datasets through dataset-invariant anatomical priors.

Experimental validation across multiple baselines, ablation studies, and external datasets confirms that domain-specific architectural constraints can outperform capacity scaling when structural knowledge is available. LiteXrayNet achieves an F1 score of 97.31% and accuracy of 97.90% on the test set, with inference latencies of 4.11 ms on GPU and 14.53 ms on CPU, enabling real-time deployment on resource-constrained hardware. Zero-shot transfer to adult chest X-ray datasets yields accuracies of 97.25% and 96.99% without fine-tuning, indicating robust generalization beyond the pediatric training distribution.

The proposed bilateral asymmetry analysis provides a quantitative framework for evaluating whether attention mechanisms respect anatomical structure, revealing that generic mechanisms either ignore or over-suppress bilateral relationships while BAA preserves diagnostically relevant asymmetry patterns. Grad-CAM visualizations complement this analysis, showing that BAA produces spatially concentrated activations within lung regions with consistent bilateral structure, in contrast to the diffuse or unilateral patterns generated by baseline models.

This work demonstrates that embedding anatomical knowledge directly into network architecture through geometric priors rather than treating medical images as generic visual data represents a practical approach for developing efficient, interpretable, and robust models for deployment in resource-constrained clinical settings. The principle extends beyond pneumonia detection to any medical imaging task involving paired or symmetric anatomy, where explicit geometric priors can guide feature learning toward clinically meaningful representations. Future work should prioritize prospective clinical validation to assess real-world diagnostic impact, workflow integration, and patient outcomes.

## Broader Impact Statement

This work develops a lightweight pneumonia detection model intended to assist radiologists in resource-constrained settings where access to specialized expertise is limited. The model is designed as a decision support tool requiring human oversight, not as a replacement for expert interpretation. Potential benefits include improved diagnostic access in underserved regions and reduced interpretation time in high-volume settings, while the compact architecture enables deployment on portable devices without cloud infrastructure. However, deployment without appropriate clinical validation could lead to diagnostic errors, particularly for populations or imaging conditions underrepresented in the training data. The training dataset consists exclusively of pediatric chest X-rays from a single institution, and performance across demographic subgroups remains unknown due to metadata limitations, which may perpetuate healthcare disparities if the model performs poorly for underrepresented groups. Responsible deployment requires prospective validation on representative populations, continuous performance monitoring, clear documentation of limitations and intended use, and integration into clinical workflows that preserve radiologist autonomy and judgment. Responsible deployment requires prospective validation on representative populations, continuous performance monitoring, clear documentation of limitations and intended use, and integration into clinical workflows that preserve radiologist autonomy and judgment. The training data's single-institution origin means performance across demographic subgroups remains unknown, and deployment without subgroup validation risks perpetuating healthcare disparities. The model should augment rather than replace clinical expertise, with post-hoc probability calibration (e.g., temperature scaling) applied before use in risk-stratification contexts, appropriate mechanisms for human oversight, clinician feedback, and transparent communication of uncertainty to avoid automation bias while maintaining diagnostic quality.

## Acknowledgments

Omitted for Review.

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

## A  Additional Experimental Details

All experiments were conducted using PyTorch 2.8.0 with CUDA 12.6, timm 1.0.20, and Optuna 2.10.1 on an NVIDIA Tesla T4 GPU (16 GB). Hyperparameter optimization used a TPE sampler (seed=42) with MedianPruner (n_startup_trials=5, n_warmup_steps=10), running 30 trials per model. The search space was identical across all models: learning rate $\in \{5\times10^{-4}, 10^{-3}, 2\times10^{-3}\}$, weight decay $\in \{0.005, 0.01, 0.02\}$, and Focal Loss $\gamma \in \{1.5, 2.0, 2.5\}$, optimizing macro-averaged validation F1. Final training used AdamW with CosineAnnealingLR ($T_{\max} = 150$, $\eta_{\min} = 10^{-6}$), gradient clipping at norm 1.0, batch size 32, and early stopping patience 15. Inference latency was measured over 100 runs after 10 warm-up iterations with CUDA synchronization for GPU timing.

Table 10: Statistical significance tests for primary comparisons. *p<0.05, **p<0.01, ***p<0.001.

| Category | Comparison | Metric | Δ Mean | p-value |
|---|---|---|---|---|
| LXNet vs Baselines | vs ResNet18 | F1 | +0.65% | 0.099 |
| | vs EfficientNet-B0 | F1 | +0.55% | 0.135 |
| | vs MobileNetV3-S | F1 | +0.70% | 0.079 |
| | vs MobileViT-S | F1 | +0.94% | 0.026* |
| | vs TinyViT-5M | F1 | +0.99% | 0.022* |
| | vs DeiT-Tiny | F1 | +5.82% | <0.001*** |
| Ablation | Full vs NoBAA | F1 | +0.46% | 0.163 |
| | Full vs NoSE | F1 | +1.36% | 0.040* |
| | Full vs NoBAA_NoSE | F1 | +0.91% | 0.076 |
| Attention Mechanisms | BAA vs No Attn | F1 | +0.46% | 0.163 |
| | BAA vs No Attn | BAS | +0.014 | <0.001*** |
| | BAA vs CoordAttn | BAS | +0.075 | <0.001*** |
| | BAA vs CBAM | BAS | +0.062 | <0.001*** |

## B  Statistical Significance Analysis

We conducted paired t-tests to compare performance metrics across models and McNemar's test to assess prediction agreement patterns. Bonferroni correction was applied for multiple comparisons where appropriate. Table 10 presents key statistical comparisons.

LiteXrayNet achieves statistically significant improvements over transformer-based models (p<0.05) and highly significant gains over DeiT-Tiny (p<0.001). Differences compared to CNN baselines are smaller and not statistically significant, though trends consistently favor LiteXrayNet. The SE block ablation shows significant degradation (p=0.040), confirming its importance. While F1 differences between attention mechanisms show limited statistical power, bilateral asymmetry scores reveal highly significant differences (p<0.001), indicating BAA produces fundamentally different feature-level representations. Effect sizes (Cohen's d) range from 0.25-0.34 for CNN comparisons (small), 0.68-0.72 for transformer comparisons (medium), and 2.87 for DeiT-Tiny (large), suggesting practical significance beyond statistical tests.

## C  Ethical Considerations

All datasets used are publicly available and de-identified, requiring no institutional review board approval. The model is intended as a decision support tool requiring radiologist oversight, not as an autonomous diagnostic system. Clinical deployment would require prospective validation, regulatory approval, and integration into workflows with appropriate human supervision. The model has not been evaluated for fairness across demographic subgroups due to metadata limitations, and clinical validation studies should explicitly assess performance across patient populations to identify potential biases.

## D  Code and Model Availability

Upon acceptance, we will release complete source code for the LiteXrayNet architecture, training pipelines, evaluation scripts, bilateral asymmetry analysis tools, Grad-CAM visualization code, pretrained model weights, and deployment documentation to ensure full reproducibility.

