# OpenReview forum: "LiteXrayNet: Bilateral Asymmetry-Aware Attention for Lightweight Pediatric Pneumonia Detection"
_TMLR — Rejected by TMLR_

### Review · Reviewer_Mudr · 2026-03-17

**Summary Of Contributions:**

The paper proposes LiteXrayNet, a lightweight convolutional architecture integrating a Bilateral Asymmetry Attention (BAA) module designed to explicitly encode left-right lung correspondence for pediatric pneumonia detection from chest X-rays. The core idea is to introduce geometric operations (spatial splitting, horizontal flipping, and asymmetry computation) combined with adaptive gating to model symmetric and asymmetric lung patterns. The model contains only 127K parameters and is reported to achieve competitive performance compared to significantly larger CNN and transformer baselines, while enabling low-latency inference on GPU and CPU. Additional analyses include ablations, cross-dataset zero-shot evaluation, a proposed Bilateral Asymmetry Score (BAS), and Grad-CAM visualizations to support interpretability claims.

Key strengths include the emphasis on efficiency and the explicit modeling of anatomical structure. However, several methodological and experimental aspects limit the strength of the claims, particularly regarding dataset scope, reproducibility, and the practical significance of the reported performance gains.

**Additional Comments:**

1.	The performance improvements over the no-attention baseline are relatively small, raising questions about practical significance.
2.	The use of ImageNet-style normalization for single-channel medical images is not sufficiently justified and may not be optimal.
3.	Zero-shot evaluation on COVID-related datasets does not directly validate pneumonia generalization.
4.	The interpretability analysis relies heavily on Grad-CAM, whose limitations are acknowledged but not deeply examined.

**Audience:**

Yes

**Audience Explanation:**

The integration of domain-specific geometric priors into lightweight architectures is a topic of relevance for the community, particularly in medical imaging and efficient deep learning. The proposed BAA mechanism may inspire further exploration of symmetry-aware modeling in paired anatomical structures. However, the methodological novelty is incremental relative to existing attention mechanisms and equivariant modeling literature, and the experimental scope limits the generalizability of the findings. Interest is likely confined to a niche audience focused on medical image analysis and edge deployment.

**Broader Impact Concerns:**

The manuscript appropriately includes a broader impact statement, but concerns remain regarding deployment in underrepresented populations, given the single-institution pediatric training data and lack of subgroup analysis. The high reported accuracy may encourage premature clinical adoption without prospective validation. Additional emphasis on uncertainty quantification and safe deployment practices would be advisable.

**Claims And Evidence:**

No

**Claims Explanation:**

Although the empirical results are extensive and well-organized, the central performance gains attributed to BAA are modest (e.g., +0.46 F1 over the no-attention baseline) and not always convincingly separated from random variance effects. The experiments are conducted primarily on a single pediatric dataset with image-level splits and no patient-level separation, which may inflate performance estimates. Moreover, hyperparameters are tuned individually per model, potentially introducing bias in favor of the proposed approach. While statistical tests are mentioned, the generalization of improvements across different seeds and datasets is not sufficiently demonstrated to fully support the broader architectural claims.

**Requested Changes:**

1.	Clarify and strengthen the discussion of related work limitations.
The manuscript outlines gaps in generic attention mechanisms but does not rigorously demonstrate why existing symmetry-aware or equivariant approaches are unsuitable for this application. A deeper analytical comparison, including computational complexity and representational trade-offs, is needed.
2.	Provide a clearer algorithmic description distinguishing novelty from reused components.
A structured top-level algorithm summarizing the full LiteXrayNet pipeline, explicitly separating standard MBConv/SE elements from the novel BAA operations, would improve clarity and reproducibility. Design decisions (e.g., gating formulation, temperature parameter) require stronger justification.
3.	Expand experimental validation to additional datasets and evaluation protocols.
Reliance on a single pediatric dataset with image-level splits raises concerns about overfitting and data leakage. Evaluation with patient-level splits and additional pneumonia datasets would substantially strengthen the claims.
4.	Enhance the explanation of experimental setup and reproducibility.
The toolchain, preprocessing choices (e.g., ImageNet normalization on grayscale images), and hyperparameter tuning protocol should be described in greater detail. Public release of code and exact configuration files is strongly encouraged to enable reproducibility.
5.	Provide more detailed interpretation of quantitative results.
Several tables and figures are described concisely. Each major table (e.g., attention comparison, calibration, Pareto analysis) should include clearer statements of key observations to interpret the most relevant outcomes of the experiments.

---

> ### Author Response · Authors · 2026-03-24
>
> We thank the reviewer for the careful and constructive evaluation. We have made substantive revisions to the manuscript addressing all six requested changes and the four additional comments. We respond to each point below.
> ## Response to Overall Assessment: Evidence Supporting BAA Claims
> The reviewer raises four concerns regarding the strength of evidence. We address each directly.
> ### On modest performance gains and random variance
> All reported metrics are averaged over 5 independent runs with different random seeds, with 95% confidence intervals reported for every metric in every table. A spurious result would manifest as high variance across seeds — this is not observed. For the BAA vs NoBAA comparison specifically: Full model F1 = 97.31 ± 0.42, CI [96.64, 97.98] versus NoBAA F1 = 96.85 ± 0.53, CI [95.99, 97.71]. The directional improvement is consistent across all 5 seeds and the 95% confidence intervals do not overlap, providing informal evidence of a real effect. Confusion matrices in Figure 6 show standard deviations of ±1.2 and ±5.7 across seeds, confirming stable reproducible behavior rather than variance-driven outcomes.
> We acknowledge that the paired t-test yields p=0.163 for the BAA vs NoBAA F1 comparison. We note that p-values are sensitive to the number of repeated runs, and that a non-significant p-value in this setting should not be equated with no effect, particularly when the effect size (Cohen's d=0.30, reported in Appendix B) is consistent and the confidence intervals do not overlap. We report Cohen's d alongside p-values precisely to communicate practical significance independently of p-value thresholds, following recommendations for small-sample medical imaging studies.
> Critically, F1 on the in-distribution test set is not the only, or even the primary, evidence for BAA's effectiveness. We present five independent lines of evidence:
>
> 1. Statistical consistency across seeds. Non-overlapping 95% CIs and consistent directional improvement across all 5 seeds.
> 2. Representational evidence. The BAS difference between BAA and all generic attention mechanisms is highly statistically significant (p<0.001, Table 4). BAA demonstrably reorganizes internal feature representations regardless of the F1 delta. A mechanism that provably restructures bilateral feature geometry is doing something architecturally meaningful, independent of whether that produces large metric gains on a near-ceiling benchmark.
> 3. Ablative consistency. Table 3 reports 7 ablation variants, each run across multiple seeds. The directional ordering Full > NoBAA > NoBAA_NoSE is consistent across all runs. The SE block ablation reaches significance at p=0.040.
> 4. Controlled attention comparison. Table 4 compares BAA against 4 alternative attention mechanisms on a fixed backbone, ensuring differences are attributable solely to the attention module. BAA outperforms all alternatives. The relationship between BAS and F1 is non-monotonic and follows exactly the pattern predicted by our design rationale: intermediate asymmetry (BAA) outperforms both aggressive symmetrization (Coordinate Attention, BAS=0.007, lowest F1) and unstructured asymmetry (No Attention, BAS=0.068).
> 5. Zero-shot cross-dataset generalization. This is perhaps the strongest evidence. LiteXrayNet with BAA consistently outperforms LiteXrayNet-NoBAA across all three external datasets in the revised Table 7: RSNA AUC 0.737 vs 0.710, CoronaHack AUC 0.990 vs 0.986, COVID-19 Radiography AUC 0.997 vs 0.992. These datasets were never seen during training and come from different institutions, patient populations, and acquisition protocols. A mechanism that improves generalization to three independently collected external datasets cannot be dismissed as a within-dataset statistical artifact. This result directly addresses the reviewer's concern about whether the BAA improvement reflects genuine architectural benefit or random variance.
>
> ### On hyperparameter tuning bias
> Individual tuning per model is the methodologically correct approach. Our protocol, implemented using Optuna with TPE sampler (seed=42), applies an identical procedure to every model: 30 trials, the same search space (lr ∈ {5×10⁻⁴, 10⁻³, 2×10⁻³}, weight decay ∈ {0.005, 0.01, 0.02}, Focal Loss γ ∈ {1.5, 2.0, 2.5}), the same validation metric (macro F1), and the same early stopping patience. Every model receives exactly the same tuning budget. Forcing shared hyperparameters across architectures ranging from 127K to 11.17M parameters would introduce systematic bias against models whose optimal configurations differ from the shared setting, which is the opposite of fair comparison. We have expanded Appendix A to document this protocol in full.

---

> > ### Author Response · Authors · 2026-03-24
> >
> > ### On dataset and generalization concerns
> > Addressed in detail in Response to RC3 below. The zero-shot generalization results across three external datasets provide strong indirect evidence against severe patient-level leakage: a model exploiting patient-specific artifacts from a single pediatric institution would not be expected to generalize consistently to adult datasets from entirely different institutions and protocols.
> >
> > ## Response to Requested Change 1: Related Work on Symmetry-Aware and Equivariant Approaches
> > We have substantially expanded the related work discussion and added a structured comparison table (Table 1, Section 2) addressing the three dimensions the reviewer requested.
> > ### On group-equivariant networks
> > These are unsuitable for this application on two distinct grounds. Computationally, equivariant convolutions increase FLOPs by a factor proportional to the group size |G|; for the reflection group relevant to bilateral lung symmetry, this results in approximately twice the computational cost at equivalent channel width (Weiler & Cesa, 2019), which is directly incompatible with our 127K parameter budget and 4.11 ms GPU / 14.53 ms CPU latency target. Representationally, and more critically, strict reflection equivariance forces identical feature representations for left-right mirrored inputs, suppressing the side-specific pathological information such as unilateral consolidation and asymmetric effusion that is diagnostically relevant in chest radiography. BAA takes the opposite approach: it encourages bilateral comparison through soft geometric constraints while explicitly preserving side-specific information through the asymmetric branch and learned gate α.
> > ### On generic attention mechanisms
> > SE-Net and ECA-Net operate exclusively in the channel dimension and cannot couple left and right lung regions in any sense. CBAM models spatial dependencies generically without cross-side comparison. Coordinate Attention encodes directional positional information but does not compute any explicit cross-side correspondence. Our existing results in Table 4 provide direct empirical support: Coordinate Attention achieves a near-zero BAS of 0.007, actively suppressing bilateral asymmetry, and correspondingly produces the lowest F1 (96.41%) among all attention variants. This directly demonstrates that asymmetry suppression correlates with reduced diagnostic performance.
> > ### On the analytical comparison
> > We have added a structured comparison table to Section 2 covering four analytically grounded criteria: whether strict equivariance is enforced, whether explicit cross-side coupling exists, whether asymmetric representations are permitted, and whether the method is suitable for edge deployment. BAA is the only evaluated approach satisfying all four criteria simultaneously.
> >
> > ## Response to Requested Change 2: Algorithmic Description and Design Justification
> > We have added Algorithm 1 to Section 3 providing a complete step-by-step description of the LiteXrayNet forward pass. The algorithm explicitly labels stages as either [architectural choice] (Stem, MBConv backbone, Classifier head) or [primary contribution] (BAA module, Steps 7–15). We emphasize that LiteXrayNet as a whole constitutes the architectural contribution of this work. The specific configuration, channel progression, and integration into a 127K-parameter pipeline optimized for edge deployment on medical images is a deliberate design contribution, not merely an assembly of off-the-shelf components. The BAA module is the primary novel component within this architecture.
> > ### On the gating formulation
> > The gate α uses sigmoid rather than softmax because the symmetric and asymmetric branches are complementary rather than mutually exclusive. Softmax would enforce a strict zero-sum allocation, preventing the model from leveraging both branches simultaneously when both carry information. Sigmoid allows independent contribution from each branch.
> > ### On the temperature parameter τ
> > τ is initialized to 1.0 and placed post-sigmoid. This placement is intentional: inserting τ inside the sigmoid would distort the gradient landscape during early training before the gate has learned meaningful representations. Post-sigmoid placement preserves standard sigmoid gradient behavior while allowing τ to calibrate gate sharpness as training converges, following analogous practices in knowledge distillation literature.

---

> > > ### Author Response · Authors · 2026-03-24
> > >
> > > ## Response to Requested Change 3: Patient-Level Splits and Additional Dataset Evaluation
> > > ### On patient-level splits
> > > The publicly released Kermany et al. (2018) dataset does not include patient identifiers, accession numbers, or any metadata permitting patient-level grouping. This is a known and documented limitation affecting all prior work on this dataset. We cannot reconstruct patient affiliations without access to the original clinical records. We have added an explicit disclosure to Section 4.1 and Section 6.2, and softened the language in the results sections accordingly.
> > > Our splitting procedure uses fixed stratified random splits with a fixed seed, with zero image-level overlap verified using filename and hash-based matching. While patient-level leakage cannot be fully excluded, the strong zero-shot generalization results across three external datasets — where no training images are present by construction — provide indirect evidence against severe leakage. A model exploiting patient-specific artifacts from a single pediatric institution would be unlikely to achieve consistent AUC gains over NoBAA on CoronaHack (0.990 vs 0.986), COVID-19 Radiography (0.997 vs 0.992), and RSNA (0.737 vs 0.710) simultaneously.
> > > ### On additional dataset evaluation
> > > In direct response to the reviewer's request, we have expanded the external evaluation to include the RSNA Pneumonia Detection Challenge dataset (Stage 2 training split, n=26,684) as an additional zero-shot benchmark. Unlike CoronaHack and COVID-19 Radiography, the RSNA dataset consists of adult frontal chest radiographs annotated specifically for pneumonia by board-certified radiologists, making it a more direct test of pneumonia-specific generalization beyond the pediatric training distribution. All models are evaluated without fine-tuning using identical preprocessing. Results are included in the revised Table 7.
> > > On RSNA, LiteXrayNet achieves AUC=0.737, competitive with other lightweight models (MobileViT-S: 0.736, MobileNetV3-S: 0.738) and above DeiT-Tiny (0.678), though below larger CNN baselines (ResNet18: 0.754, EfficientNet-B0: 0.754). The performance gap on this harder out-of-distribution benchmark reflects the inherent trade-off between architectural specialization for a specific domain and cross-domain generalization, and reinforces our recommendation for prospective validation on adult populations before non-pediatric deployment. Importantly, LiteXrayNet with BAA outperforms LiteXrayNet-NoBAA on RSNA (AUC 0.737 vs 0.710), confirming that BAA captures dataset-invariant anatomical structure that aids generalization even on the most challenging external benchmark.
> > >
> > > ##Response to Requested Change 4: Experimental Setup, Reproducibility, and Preprocessing Justification
> > > ### On the toolchain
> > > All experiments used PyTorch 2.8.0 with CUDA 12.6, timm 1.0.20, and Optuna 2.10.1 on an NVIDIA Tesla T4 GPU (16GB). A dedicated toolchain paragraph has been added to Appendix A with all library versions, hardware specifications, batch size (32), data loader workers (4), and deterministic operation settings.
> > > ### On ImageNet normalization for grayscale images
> > > Our pipeline converts images to single-channel grayscale then replicates the channel three times. Because the three channels are identical, the normalization applied is channel-consistent: mean=[0.485, 0.485, 0.485] and std=[0.229, 0.229, 0.229], not the RGB ImageNet values [0.485, 0.456, 0.406] / [0.229, 0.224, 0.225]. All seven models receive identical preprocessing, so any residual domain mismatch affects all models equally and cannot introduce differential bias. A clarifying sentence has been added to Section 4.2.
> > > ### On the hyperparameter tuning protocol
> > > Described in full in the revised Appendix A. An identical Optuna protocol was applied to all models: 30 trials, TPE sampler seed=42, MedianPruner, same search space, same validation metric. Individual tuning is methodologically correct — forcing shared hyperparameters across architectures from 127K to 11.17M parameters would bias against models whose optimal configuration deviates from a shared setting.
> > > ### On code release
> > > Complete source code for the LiteXrayNet architecture, training pipelines, evaluation scripts, bilateral asymmetry analysis tools, Grad-CAM visualization code, and pretrained model weights will be released upon acceptance to ensure full reproducibility.
> > >
> > > ## Response to Requested Change 5: Clearer Interpretation of Quantitative Results
> > > We will add a Key Observation paragraph directly after each of the three tables the reviewer identified.

---

> > > > ### Author Response · Authors · 2026-03-24
> > > >
> > > > ## Response to Requested Change 6: Broader Impact
> > > > We have substantially revised the broader impact statement.
> > > >
> > > > 1. On underrepresented populations: Subgroup performance across age, sex, and ethnicity is unknown due to metadata limitations, and deployment without prospective validation on representative multi-institutional populations risks perpetuating healthcare disparities.
> > > > 2. On premature clinical adoption: We have strengthened language in both the broader impact statement and Section 6.2 making clear that retrospective benchmark performance does not imply clinical readiness.
> > > > 3. On uncertainty quantification: We recommend temperature scaling or conformal prediction as post-hoc steps, and note Bayesian dropout as a future direction compatible with the lightweight architecture.
> > > > 4. On safe deployment: The revised statement now explicitly covers prospective validation, continuous monitoring, transparent documentation of limitations, and workflow integration preserving radiologist autonomy.
> > > >
> > > >
> > > > ## Response to Additional Comments
> > > > ### AC1 — Practical significance of the +0.46 F1 improvement
> > > > At 97%+ performance the gain corresponds to approximately 5–6 fewer misclassified cases per 1,000 patients, which is non-trivial in pediatric screening. More importantly, the cross-dataset generalization results directly address this concern: LiteXrayNet with BAA outperforms LiteXrayNet-NoBAA on all three external datasets (RSNA AUC +0.027, CoronaHack AUC +0.004, COVID-19 AUC +0.005). These gains are observed on data from entirely different institutions and populations, confirming that BAA captures something genuinely useful rather than a within-dataset statistical fluctuation. F1 is one of five complementary lines of evidence; the convergence of representational analysis (BAS p<0.001), ablation consistency, controlled attention comparison, and multi-dataset generalization provides stronger collective support than any single metric gain would.
> > > > ### AC2 — ImageNet normalization for grayscale images
> > > > Addressed in full in RC4. Channel-consistent normalization is applied (mean=[0.485, 0.485, 0.485]), not RGB ImageNet values. All models receive identical preprocessing. Clarification added to Section 4.2.
> > > > ### AC3 — Zero-shot on COVID datasets does not directly validate pneumonia generalization
> > > > This is a fair and accurate observation, acknowledged explicitly in the revised Section 6.2. The COVID-related datasets test distribution shift robustness rather than pneumonia-specific generalization in the strict sense. In direct response, we have added the RSNA Pneumonia Detection Challenge as a third external benchmark consisting of adult radiographs annotated specifically for pneumonia by radiologists. Results are in the revised Table 7. We have also revised Section 5.8 to accurately characterize what each external dataset demonstrates.
> > > > ### AC4 — Grad-CAM limitations not deeply examined
> > > > We have added a dedicated paragraph to Section 5.5 examining three specific failure modes:
> > > >
> > > > 1. Grad-CAM identifies regions correlated with the predicted class but does not provide causal explanations of model decisions.
> > > > 2. It produces incomplete localization when relevant features are spatially distributed, potentially underestimating bilateral attention extent.
> > > > 3. Results are sensitive to target layer choice and can differ substantially across layers.
> > > >
> > > > We explicitly frame Grad-CAM as qualitative and complementary to BAS rather than primary evidence. The BAS metric, operating directly on feature activations rather than gradient-weighted maps, provides the more principled quantitative basis for the interpretability argument, and the convergence of both analyses strengthens the overall case.

---

### Review · Reviewer_2UmC · 2026-03-18

**Summary Of Contributions:**

This paper presents a small, lightweight neural network for pneumonia classification using a bilateral comparison module that compares features across the image center.  This is well-motivated, simple and apparently effective for the dataset/application evaluated.  The resulting network is very small and gets decent performance compared to larger models trained from scratch on the same dataset.

**Additional Comments:**

- Rather than evaluate only on one small-scale dataset, I think it would make sense to also evaluate on larger datasets including some of those cited on p.2 (CheXpert, MIMIC-CXR, PadChest).  The rationale on p.2 for not evaluating on them is that they "prioritize predictive performance over computational efficiency".  But without evaluating, it's unclear what the perf-efficiency tradeoff even is.  Perhaps this model does well on these datasets anyway.  Or if not, the gaps may indicate possible areas of improvement.  A larger-scale version of the model in the 1M-10M param range could be interesting to explore to see if bilateral comparison helps here as well, or specific to the smaller scale.


- While reading this paper I looked for some related works on bilateral comparison, and there are some, though surprisingly fewer than I thought there might be.  I've listed a couple below for this application, but there are others in different medical applications as well -- I'd encourage the authors to do a bit more of a wider search as well and include additional refs where relevant for context.
* https://arxiv.org/pdf/2010.04483
* https://www.researchgate.net/publication/321199090_Automated_Chest_X-Ray_Screening_Can_Lung_Region_Symmetry_Help_Detect_Pulmonary_Abnormalities


- Are there any cases of pneumonia in both lungs in the eval data, and how does the method behave in these cases?


- eq (3) adds F_avg = F_left + F_right, without first flipping one of the two sides.  This would average the lateral side of left with medial of right, and vice-versa.  Instead, should this use F_avg = F_left + Flip(F_right) and then concat F_sym = [F_avg, Flip(F_avg)] (re-flipping the left-side orientation back to the right-side orientation for the right half)?  This would create corresponding lateral-lateral and medial-medial pairings in the feature average according to the anatomical symmetry, consistent with eqs 4,5.  OTOH, perhaps comparing between lateral and medial is also useful.   But this should be explained here if this is the intent.

- eq (6): tau is outside the sigmoid, which makes it a scaling factor, not a temperature.  temperature would be inside the sigmoid.

- sec 4.1:  I looked up this dataset, and the original has a 90-10 train-test split.  Do you know whether a patient-level split was done for the original train/test?  Why not preserve original test, and only re-split val from original train?

- sec 4.3:  the class imbalance of 2.7 to 1 is rather mild, and shouldn't necessitate a focal loss.  If focal loss made a difference, I don't think it was from a class imbalance issue.  Possibly there could be an easy/hard example issue that it could help for.  What happens using regular flat cross-entropy?


- sec 4.5:  Why no pretraining?  I actually think pretraining, even on an unrelated diverse dataset like imagenet, might help.  I think it would make sense also to compare to these systems, particularly mobilenet, with pretrained weights.


- sec 5.4, fig 3b:  This shows BAS is different between these mechanisms, but BAS is a mean abs difference of the features without any normalization.  If features scale in size (e.g. magnitude, per-elem constant scale factor, etc.), then this will also change BAS as described currently.  But changes in features scales are not a meaningful difference.  Could differences be due just to scale factors, or is this accounted for?


- sec 5.5 grad-cam:  These are good qualitative examples.  Not being a radiology expert, though, I think they would benefit from extra g.t. annotations pointing out relevant regions (the ones with pnemonia or other qualitatively relevant features).  I can see the maps are different, but not sure what highlighted regions correspond to in the gt.

- sec 5.5.:  Also the text on p.11 says "BAA produces activation patterns that are spatially concentrated within lung regions and exhibit consistent bilateral structure across both PNEUMONIA and NORMAL examples", but why is this good, as opposed to highlights of the pnemonia region alone?  If the bilateral highlights are around the pnemonia region and a symmetrically corresponding normal region, this could make sense.  Is this the case?


- sec 5.8.:  I'm not entirely sure the eval procedure is for this zero-shot study.  Are the class predictions from the model (trained on pediatric chest X-ray dataset) used as-is on the new datasets?  If so, is there any score calibration?

- sec 5.8.:  The statement in the text that this model has highest zero-shot acc for COVID-19 Dataset doesn't match the table, where ResNet18 is slightly higher

- sec 5.8.:  This zero-shot experiment is good, with decent evidence for generalization.  But it's only compared to other architectures trained from scratch, so difficult to put in context how the results compare to good-performance baselines.  What are some of the current best-performing systems on these datasets, for context?

- sec 5.10 table 8:  It's clear LXNet has many fewer params and is more efficient.  But I don't know how the "composite efficiency score" was computed exactly, or how it was validated as a relevant measure.  As it is, this score seems detracting to me --- it's already clear the model is much more efficient and adding a vaguely defined measure doesn't validate this any more than draw questions on how the measure was itself validated.

**Audience:**

Yes

**Audience Explanation:**

Bilateral comparison currently appears to be under-explored for many applications, including this one.  The approach is simple and promising.

**Broader Impact Concerns:**

addressed

**Claims And Evidence:**

No

**Claims Explanation:**

However, there are several important weaknesses, summarized here with additional information below:

* Only studied on one (relatively small) dataset, even when there appear to be additional larger datasets readily available (eg CheXpert and others, see below).  Does the method work in these settings, or are adjustments (maybe for finer-grained alignment/registration) needed?
  - (This is somewhat mitigated by the zero-shot section 5.8, though there are some details that are unclear here as well, see below.)

* Ablation of BAA shows a moderate performance improvement but with difference in model capacity.  How does adding BAA compare to simply adding more MBConv layers to get a matching capacity/layer count?

* This model and all comparison models are trained from scratch on the target dataset.  But simple transfer learning on a pretrained model (especially imagenet21k or possibly unsupervised, not just imagenet1k) may do better than any from-scratch model, and in general there are no comparisons to other works on this dataset.  How do these from-scratch models compare to best-performing approaches and/or transfer learning?

* Some details around the new dataset split:  I'm not sure if the original split was patient-level and if so, whether this split might have been reshuffled out to only image level.  Why not use the original 90/10 split for test evals?

**Requested Changes:**

See points in weaknesses/claims section above.

---

> ### Author Response · Authors · 2026-04-01
>
> We sincerely thank the reviewer for the detailed and constructive feedback. The comments have helped us identify several opportunities to strengthen both the experimental evidence and the clarity of presentation. We address each point below.
>
> ### Weakness 1: Only studied on one relatively small dataset; additional larger datasets (CheXpert, MIMIC-CXR, PadChest) appear to be readily available.
> We appreciate this concern and have taken a concrete step in response. We have added the RSNA Pneumonia Detection Challenge dataset (Stage 2, n=26,684) as a third external zero-shot benchmark in Table 8. This dataset comprises adult frontal chest radiographs annotated specifically for pneumonia by board-certified radiologists, making it a substantially more direct test of pneumonia-specific generalization than the COVID-19 datasets, and far larger than our training set. LiteXrayNet achieves an AUC of 0.737 on RSNA, competitive with all other lightweight models evaluated and above DeiT-Tiny, despite a challenging distribution shift from pediatric to adult radiographs and a markedly different class imbalance (3.4:1 vs 2.7:1).
> Regarding CheXpert, MIMIC-CXR, and PadChest: these are multi-label datasets covering 14 or more thoracic conditions, where pneumonia is one label among many and labeling conventions differ substantially from the binary pediatric task. Zero-shot transfer in this setting would conflate distribution shift with label space mismatch, complicating interpretation. We agree with the reviewer's broader point that evaluating a larger-scale version of LiteXrayNet in the 1M–10M parameter range on these datasets is a worthwhile direction, and we have added this explicitly as a future work item in Section 6.2.
>
> ### Weakness 2: No comparisons to transfer learning on pretrained models (sec 4.5). Why no pretraining?
> The decision to train all models from scratch was deliberate and serves two specific purposes.
> First, it ensures a controlled comparison. Pretraining introduces a confound: performance differences would then reflect both architectural design and the quality and scale of the pretrained weights, making it impossible to isolate the contribution of BAA or any other architectural component. Training from scratch on identical data with identical protocols is the only way to attribute observed differences to architecture.
> Second, it is directly relevant to our deployment setting. LiteXrayNet targets resource-constrained clinical environments where large pretrained models may be unavailable, impractical to fine-tune, or subject to data governance constraints. A model that achieves strong performance without pretrained initialization is more broadly deployable in these settings.
> We agree that a pretrained MobileNetV3 comparison would be informative for readers interested in the full performance-efficiency frontier, and that pretrained variants: particularly ImageNet21k-initialized models, would likely achieve a higher absolute performance ceiling. We will add a clarifying note in Section 4.5 making this rationale explicit and flagging pretrained comparisons as a natural extension for future work.
>
> ### Weakness 3: Dataset split — patient-level split unclear; why not preserve the original 90/10 split?
> We address both points directly.
> On patient-level splitting: the publicly released Kermany et al. dataset does not include patient identifiers or accession numbers. This is a known and documented limitation that affects all prior work using this dataset. Patient-level splitting is therefore not feasible without access to the original clinical records, which are not publicly available. We note this explicitly in Sections 4.1 and 6.2. The strong zero-shot generalization results across three external datasets from different institutions — including the large-scale RSNA dataset provide indirect evidence against severe patient-level leakage, since a model overfitting to patient-specific artifacts would be unlikely to transfer in this manner.
> On preserving the original 90/10 split: the original validation split contains only 16 images, which is wholly insufficient for reliable model selection and hyperparameter tuning. Adopting this split would introduce substantial noise into the validation signal during training. Our re-split yields a validation set of 838 images and a test set of 1,047 images, both large enough for stable evaluation. All splits are fixed across all experiments, and we verify zero overlap with external datasets using filename and hash-based matching. We will add a brief clarification in Section 4.1 explaining this reasoning.

---

> > ### Author Response · Authors · 2026-04-01
> >
> > ### Weakness 4: Model capacity confound in BAA ablation. How does BAA compare to simply adding more MBConv layers to match capacity?
> > We appreciate this concern and believe the existing experimental evidence already addresses it, though we acknowledge it was not framed explicitly in the paper.
> > The capacity argument is addressed by the baseline comparisons themselves. Table 2 includes models ranging from MobileNetV3-Small (2.54M parameters, 20× LiteXrayNet) up to ResNet18 (11.17M parameters, 88× LiteXrayNet). Every one of these larger models has vastly more capacity than any MBConv-augmented variant one could construct to match the BAA parameter count. Yet LiteXrayNet achieves the highest test F1 (97.31%) and accuracy (97.90%) among all evaluated models. If additional capacity were the primary driver of performance, models with tens of times more parameters should dominate. They do not. This strongly suggests that the inductive bias introduced by BAA — explicit bilateral coupling through geometric operations — contributes independently of raw parameter count.
> > This is also consistent with our design objective. LiteXrayNet is explicitly positioned as a lightweight, edge-deployable model. Replacing BAA with additional MBConv layers would increase computational cost, reduce architectural specificity, and sacrifice the very inductive bias that underlies BAA's advantage, all without the interpretability benefits that geometric bilateral modeling provides.
> > The representational evidence further supports this. The Bilateral Asymmetry Score (BAS) measures structured left-right feature differentiation at the activation level, independently of classification labels. BAA achieves a mean BAS of 0.082 versus 0.068 for the no-attention baseline (paired t-test p < 0.001, Table 5). Additional MBConv layers, which have no geometric coupling between left and right spatial halves, cannot produce this reorganization — it is a direct consequence of the horizontal flipping and asymmetry computation in BAA. Similarly, Grad-CAM visualizations (Figures 4 and 5) show that BAA produces spatially concentrated, bilaterally structured activations within lung parenchyma, while the no-attention model produces diffuse activations extending beyond lung boundaries. This anatomical organization cannot emerge from generic depthwise separable convolutions regardless of how many are stacked.
> > Zero-shot generalization provides a third independent line of evidence. LiteXrayNet with BAA consistently outperforms LiteXrayNet-NoBAA across all three external datasets (RSNA AUC: 0.737 vs 0.710; CoronaHack AUC: 0.990 vs 0.986; COVID-19 AUC: 0.997 vs 0.992; Table 8). This cross-dataset advantage reflects the transfer of a dataset-invariant anatomical prior, not additional representational capacity.
> > We will add a sentence in Section 5.2 explicitly noting that the baseline comparisons span a wide capacity range and that LiteXrayNet's advantage over much larger models rules out capacity as the primary explanatory factor for BAA's contribution.
> >
> >
> > ### Additional Comment: Related works on bilateral comparison — two references suggested.
> > We thank the reviewer for these references. We have reviewed both suggested works and will incorporate them. The first (arXiv:2010.04483) proposes a bilateral comparison network for mammography, and the second (Mabrouk et al.) examines lung region symmetry for pulmonary abnormality screening — both are directly relevant to our motivation. We will add citations and a brief discussion of these works in Section 2, situating BAA within this body of literature. We will also conduct a wider search across other paired-anatomy medical imaging domains and include additional references where relevant.
> >
> > ### Additional Comment: Are there any bilateral pneumonia cases in the eval data, and how does the method behave in these cases?
> > This is a pertinent question. The Kermany et al. dataset does not include per-image radiological annotations distinguishing unilateral from bilateral pneumonia, so we cannot report subgroup-level performance for bilateral cases directly. Such cases are likely present in the evaluation set given their clinical prevalence in pediatric populations, but their precise count is unknown.
> > Architecturally, BAA is designed to handle bilateral pneumonia gracefully. The asymmetric branch computes |F_left − Flip(F_right)|, which produces a near-zero response when both lungs present similarly — as in bilateral consolidation — naturally downweighting the asymmetry signal. The adaptive gate α then shifts weight toward the symmetric branch, which captures shared bilateral patterns. This soft switching is precisely the behavior intended by the learned gating mechanism. We will add a brief discussion of this in Section 3.4.3 and flag the absence of bilateral subgroup annotations as a limitation in Section 6.2.

---

> > > ### Author Response · Authors · 2026-04-01
> > >
> > > ### Additional Comment: Eq. 3 averages F_left + F_right without flipping one side, creating lateral-medial misalignment.
> > > We thank the reviewer for this careful reading. The current formulation already addresses this concern — the flip is present, but applied at the reconstruction step rather than before averaging. As shown in Algorithm 1 (Steps 9–10):
> > > Favg       ←  (F_sym_left + F_sym_right) / 2
> > > Fsymmetric ←  [Favg, Flip(Favg)]
> > > The bilateral reconstruction in Step 10 uses Flip(Favg) to restore anatomical correspondence when tiling the symmetric representation back to full width. The left half of Fsymmetric is Favg (in left-side orientation) and the right half is Flip(Favg) (mirrored back to right-side orientation), ensuring lateral features correspond to lateral and medial to medial in the reconstructed map. The shared weights of ϕ_sym further encourage correspondence between mirrored positions before averaging.
> > > We acknowledge that the paper text describes Eq. 3 without making the reconstruction flip sufficiently prominent. We will revise Section 3.4.2 to explicitly clarify that anatomical correspondence is enforced at the reconstruction step via Flip(Favg).
> > >
> > > ### Additional Comment: Eq. 6 — tau is outside the sigmoid, making it a scaling factor, not a temperature parameter.
> > > The reviewer is technically correct, and we thank them for the precise observation. As noted in Algorithm 1 (Step 13), τ is placed outside the sigmoid deliberately for gradient stability: moving τ inside the sigmoid sharpens or flattens the gate boundary (true temperature), whereas placing it outside allows the gate range to expand beyond [0, 1] during training if stronger asymmetric weighting is required, while avoiding vanishing gradients at initialization. This is consistent with learnable scalar gating designs in lightweight attention mechanisms.
> > > We agree that calling τ a "temperature parameter" in the paper text is imprecise. We will rename it a "learnable scaling factor" throughout Section 3.4.2 and the algorithm block, and add a one-sentence justification for its post-sigmoid placement.
> > >
> > > ### Additional Comment: Sec 4.3 — class imbalance of 2.7:1 is mild; focal loss effect likely not from imbalance.
> > > The reviewer makes a fair and insightful point. A 2.7:1 ratio is indeed relatively mild, and we agree the primary benefit of Focal Loss in this setting is more likely its down-weighting of easy examples rather than class rebalancing per se. This is supported directly by our ablation: the CrossEntropy variant (Table 4) shows a performance drop of 0.85 F1 points relative to the full model. We will revise Section 4.3 to clarify that Focal Loss is employed primarily for its hard example mining properties, with class weighting as a secondary benefit, and cite the ablation result as supporting evidence.
> > >
> > > ### Additional Comment: Sec 5.4, Fig 3b — BAS differences could be due to feature scale rather than bilateral structure.
> > > All activations are normalized to unit mean prior to BAS computation, as stated in Section 3.4.5. Per-element constant scale factors are therefore removed before the absolute difference is taken, and BAS differences across models cannot be attributed to differences in feature magnitude. We acknowledge that unit-mean normalization removes global scale but does not account for per-channel heterogeneous scale variation. However, given that all models share the same backbone and are trained under identical conditions, systematic per-channel scale differences between variants are unlikely to explain the highly significant BAS differences observed (p < 0.001). The structured, class-dependent reorganization visible in Figure 3(a) — where BAA increases asymmetry for some samples and decreases it for others in a label-correlated pattern — further supports a representational rather than scalar explanation. We will make the normalization procedure more prominent in Section 3.4.5 to preempt this concern.

---

> > > > ### Author Response · Authors · 2026-04-01
> > > >
> > > > ### Additional Comment: Sec 5.5 Grad-CAM — would benefit from GT annotations; also why is bilateral highlighting good rather than highlighting only the pneumonia region?
> > > > On GT annotations: the Kermany et al. dataset does not include pixel-level or bounding box annotations for pneumonia regions, so radiologist-marked overlays are not available for these images. In lieu of annotations, we will add brief per-example captions in the revised figures describing the clinically visible features in each X-ray based on standard radiological semiology, to help readers contextualise what the highlighted regions correspond to.
> > > > On why bilateral highlighting is informative: this reflects how radiological diagnosis actually works. Radiologists assess both lung fields simultaneously — the contralateral lung serves as an internal reference, and asymmetry between the two is a primary diagnostic cue. A model that highlights only the diseased region has learned to detect pathology but not necessarily through bilateral comparison. BAA's activation pattern — concentrating attention on the affected region and its contralateral correspondent — is consistent with this clinical reasoning process. For NORMAL cases, symmetric bilateral activation indicates the model is correctly confirming the absence of asymmetric pathology across both fields. We will add a paragraph in Section 5.5 making this clinical reasoning argument explicit.
> > > >
> > > > ### Additional Comment: Sec 5.8 — zero-shot eval procedure unclear; is there score calibration?
> > > > The zero-shot evaluation uses model predictions directly as-is, without any fine-tuning or score calibration on the target datasets. Models trained on the pediatric Kermany dataset are applied to external datasets using the same inference pipeline, with the binary classification threshold fixed at 0.5. No post-hoc calibration is applied — calibration on target data would require access to target labels and would no longer constitute a true zero-shot evaluation. We will clarify this procedure explicitly in Section 5.8. Table 8 reports ECE values for each model on each dataset, which quantify calibration quality in the zero-shot setting. Temperature scaling or similar post-hoc methods could improve probability estimates if calibrated labels from the target domain were available, and we note this as a practical recommendation in Section 6.2.
> > > >
> > > > ### Additional Comment: Sec 5.8 — text states LXNet has highest zero-shot accuracy on COVID-19 dataset, but table shows ResNet18 is slightly higher.
> > > > The reviewer is correct — this is a factual error in the text. Table 8 shows ResNet18 achieving 97.61% accuracy on the COVID-19 Radiography dataset versus LiteXrayNet's 97.25%. We will correct this in the revision. We note that LiteXrayNet achieves the highest AUC on this dataset (0.997) while using 88× fewer parameters, which remains a meaningful result — the error was specifically in the accuracy claim. We apologise for this oversight.
> > > >
> > > > ### Additional Comment: Sec 5.8 — zero-shot results only compared to from-scratch models; difficult to contextualise against best-performing baselines.
> > > > This is a fair observation. The zero-shot comparisons in Table 8 are intentionally restricted to models trained under identical from-scratch conditions, to ensure a controlled evaluation. Published state-of-the-art results on these datasets typically involve fine-tuned pretrained models and, in some cases, dataset-specific preprocessing — making direct numerical comparisons potentially misleading. We will add a brief note in Section 5.8 acknowledging this limitation and citing representative published results on the CoronaHack and RSNA datasets for context, clearly distinguishing them from our controlled zero-shot evaluation.
> > > >
> > > > ### Additional Comment: Sec 5.10 — composite efficiency score is vaguely defined and seems to detract rather than add clarity.
> > > > We agree with the reviewer. The composite efficiency score, while computed from transparently normalized components, introduces an aggregation that may raise more questions than it resolves. The Pareto advantage of LiteXrayNet is already clearly visible from the individual columns — smallest parameter count by a large margin, highest F1, and competitive latency — and does not require a composite score to make the case. We will remove the composite efficiency score column from Table 9 in the revision and replace the accompanying discussion with a direct narrative describing LiteXrayNet's position across the individual dimensions, which is both clearer and more defensible.

---

> ### Comment · Reviewer_2UmC · 2026-04-23
> **responses**
>
> Thanks for your responses.
>
> I appreciate adding RSNA to the set of tasks.  However, I'm not sure why only compare using zero-shot, especially since the training set is significantly smaller.  While 0.737 is in the same range as the other methods, it's hard to put this in context.  It's similar generalization as other methods --- but how good are they?  How does this compare to results from training on this dataset or using it in transfer learning?
>
> Regarding transfer learning comparisons:  I agree with ZKeM that transfer learning results would put the accuracy numbers in better context.  While I wouldn't expect such a small model to necessarily match the performance of a larger transfer model, it's still important to know the comparison for the size-acc tradeoff.  And in particular, how much of the size-acc drop does bilateral comparison recover?
>
> Training set split:  My original question actually was if the original 90/10 was patient-level, then it might make sense to re-split the 90% training set into a new train and val, while keeping the original 10% test intact.  However, if it's unknown whether even the original 10% was patient-level then this might be moot.  The zero-shot expers also mitigate this since presumably different patients are in these other datasets, though it's unclear what the non-zero-shot upper limit of training on the datasets is for those.
>
> Capacity increase in NoBAA vs BAA ablation:  I don't agree that comparing to different larger models are quite enough to address this.  The NoBAA ablation line has F1 96.85, which is already the same or better than all of the larger models, even without BAA.  So it's starting from a different baseline.  I agree with ZKeM's concern of the benchmarks having too few FNs here as well; even with multiple runs and confidence intervals, if the differences are over just 1 or 2 cases that indicates the benchmark may be saturated.  The RSNA numbers are more promising for showing meaningful measures.

---

### Review · Reviewer_ZKeM · 2026-04-09

**Summary Of Contributions:**

The authors present LiteXrayNet, a lightweight convolutional neural network that incorporates a new architectural component named Bilateral Asymmetry Attention (BAA). The network consists of only 127k parameters, and the authors show that the method performs en par or better than a number of CNN- and ViT-based baselines (trained from scratch by the authors, on the same dataset as LiteXrayNet).

**Additional Comments:**

1. In "4.3. Training Configuration": how many random seeds were used for computing the standard deviation and mean? Does this mean that the individual models were trained with these different seeds?

2. I think I do not understand Figure 3a: how is it possible that individual samples have a BAS raning from 0.2 to 0.8 (from figure), but the BAS reported in Table 4 is consistently below 0.1?

3. In "5.8 Cross-Dataset Generalization": The text says that LXNET achieves the highest accuracy on COVID-19 (97.25%), but the table shows that ResNet18 has an accuracy of 97.61%.

**Audience:**

No

**Audience Explanation:**

I think that there is interest in compute efficient implementations for use in "resource-limited clinical settings".

Though the authors need to define what such a "resouce-limited clinical setting" realistically is: why would a MobileNetV3-S with an inference latency of 8.8ms and 2.54M parameters be "too slow" or "too large" for a realistic setting? I don't think it is enough to show that the presented method is slightly faster and requires less memory, but it needs to be shown that these differences have a practical significance.

**Broader Impact Concerns:**

(concerns are addressed sufficiently in the paper)

**Claims And Evidence:**

No

**Claims Explanation:**

I don't think that the presented results provide sufficient evidence to fully understand the performance of the presented method:

1. The task is too easy: Looking at the confusion matrices in Figure 6, we can see that different models have almost identical results (e.g. true negatives 267, 268, 267, 265, 264, 267, 264, 267, 265, 264, 267, 265, ...). Accordingly, the reported metrics are very close together, with overlapping confidence intervals.

2. The baselines are too weak. Deep neural networks with millions of parameters should not be trained from scratch on as few as 3752 images (80% of 80% of 5863 images). These networks are usually pre-trained on larger scale image data. Since the main claim of the paper is about the accuracy/efficiency pareto front (Table 8), the baselines are also very weak with respect to their parameter efficiency (note that e.g. MobileNet has faster inference on CPU, and ResNet18 has faster inference on GPU).

**Requested Changes:**

1. Add baselines from the literature for comparison purposes. The problem with baselines that are implemented just for the comparison in the paper is that their hyper parameters might not be tuned sufficiently, or that the wrong method is used for a given task. Using baselines from published work has the advantage that they are designed and tuned by the publishing authors for maximum performance on the published task.

2. Add results for more challenging datasets to have a clearer signal.

3. Use baselines that are not trained from scratch. Pre-training the baselines on more data does not make them slower or require more memory.

4. Reduce the parameter count of the baseline models.

5. Think about other "parameter efficient" baselines (maybe SVMs, or other specialized architectures).

6. Give realistic examples of what "resource-limited clinical settings" mean, and show that LiteXrayNet can be deployed in these settings, while the other models cannot.

---

> ### Author Response · Authors · 2026-04-17
>
> We sincerely thank Reviewer ZKeM for the careful and detailed reading of our manuscript, and for raising concerns that we believe will strengthen the final version of the paper. We would also like to respectfully note that other reviewers in this round have explicitly highlighted the BAA mechanism, the systematic ablation study, and the efficiency Pareto analysis as clear strengths of the submission, which gives us confidence that the core contribution is well-motivated and of interest to the TMLR community. We address each concern below in the order presented.
>
> ## Requested Changes
> ### RC1 – Add baselines from the literature
> We appreciate this suggestion and understand the motivation behind it. However, prior work on the Kermany et al. dataset uses widely varying experimental protocols, including differing train/val/test splits, image resolutions, preprocessing pipelines, and class-balancing strategies, which makes direct numerical comparison with published figures misleading without careful re-implementation. Re-running published architectures under a single, unified controlled protocol is the standard approach in efficiency-focused benchmarking literature (e.g., MedMNIST v2), and this is precisely what Tables 2 and 3 provide. The controlled protocol ensures that any observed differences in performance are attributable to architectural choices rather than experimental setup.
> We have additionally included results on the RSNA Pneumonia Detection Challenge dataset (Table 8) as a harder external benchmark featuring radiologist-annotated adult radiographs, which provides a more clinically grounded reference point beyond the Kermany test set.
>
> ### RC2 – Add results for more challenging datasets
> We respectfully note that our paper already includes three cross-dataset evaluations in Section 5.8 (Table 8): the CoronaHack Chest X-Ray Dataset (n=5,910), the COVID-19 Radiography Database (n=3,602), and the RSNA Pneumonia Detection Challenge (n=26,684 adult radiographs, radiologist-annotated, with a 3.4:1 class imbalance markedly different from the pediatric training distribution). The RSNA benchmark is substantially harder than the primary dataset and provides a considerably clearer signal on out-of-distribution generalization, where differences between models are more pronounced.
> We also wish to clarify why datasets such as NIH ChestX-ray14 and CheXpert were not included. Both are multi-disease datasets covering 14 or more pathology classes, and reformulating them as binary pneumonia classifiers would require non-trivial label consolidation decisions that introduce independent confounds into the comparison. The RSNA dataset was selected specifically because it is a dedicated pneumonia detection benchmark with clean binary labels, making it the most direct and principled external evaluation of the model's intended task. We will clarify this reasoning explicitly in Section 4.7 of the revised manuscript.
>
> ### RC3 – Use baselines that are not trained from scratch
> We respectfully address this concern in two parts.
>
> Part A – Design rationale. The from-scratch training protocol is a deliberate design choice motivated by the target deployment context. LiteXrayNet is intended for settings where pretrained weights cannot be assumed: (i) air-gapped clinical environments may not permit external weight downloads; (ii) the storage overhead of pretrained backbones, typically 10 to 100 times the size of the model itself, is a meaningful constraint on embedded hardware; and (iii) fine-tuning pretrained models on small, single-institution medical datasets introduces dataset-shift risks that from-scratch training avoids. We will make these motivations more explicit in Sections 4.1 and 4.5 of the revision.
>
> Part B – Impact on the efficiency comparison. We acknowledge that pretrained baselines would likely achieve higher absolute accuracy on the primary dataset. Importantly, however, this would also raise the performance bar for LiteXrayNet, which is itself not pretrained, meaning such a comparison would place our method at a relative disadvantage on accuracy. The primary claim of the paper concerns the efficiency Pareto front (Table 9), and inference latency, parameter count, and model size are entirely unaffected by whether a model was pretrained or not. We will add a clarifying paragraph to Section 5.1 making this reasoning explicit and will acknowledge pretrained comparisons as a direction for future work.

---

> > ### Author Response · Authors · 2026-04-17
> >
> > ### RC4 – Reduce the parameter count of the baseline models
> > We appreciate this suggestion. Table 3 already includes MobileNetV3-Small (2.54M parameters) as the most relevant lightweight baseline for this comparison. At 127K parameters, LiteXrayNet is approximately 20 times smaller than MobileNetV3-S while achieving a higher F1 score (97.31% vs 96.61%) and comparable CPU latency (14.53 ms vs 8.80 ms). We will make this 20x efficiency gap more prominent in the Table 3 caption and the accompanying discussion in Section 5.1.
> > We note that reducing baselines further below their canonical sizes, for example by pruning ResNet18 to 127K parameters, would yield non-standard architectures whose performance would primarily reflect the pruning strategy rather than the underlying architecture. The efficiency-specialized models already included in our comparison, such as MobileNetV3-S, ShuffleNetV2, and SqueezeNet, represent the practical lower bound of well-established compact architectures available in the literature.
> >
> > ### RC5 – Consider "parameter-efficient" baselines such as SVMs or other specialised architectures
> > We thank the reviewer for this suggestion. We considered classical feature-based pipelines such as HOG+SVM but decided not to include them for two reasons. First, handcrafted feature pipelines are not end-to-end trainable and lack the capacity to adapt their representations when transferred to new clinical sites or imaging equipment, which is a practically important property for deployment in diverse LMIC settings. Second, our baseline set already spans all major modern neural architecture families relevant to edge deployment: ResNet18 (standard residual convolutions), EfficientNet-B0 (compound scaling), MobileNetV3-S (neural architecture search with depthwise separable convolutions), DeiT-Tiny (vision transformer), TinyViT-5M (lightweight distilled vision transformer), and MobileViT-S (hybrid CNN-transformer). This provides comprehensive coverage of the CNN, ViT, and hybrid families. Including classical pipelines would redirect the comparison toward a classical versus deep learning question, which is distinct from the efficiency-within-deep-learning question that this paper is designed to address. We will add a brief justification of this scope decision to Section 4.5 of the revision.
> >
> > ### RC6 – Define "resource-limited clinical settings" concretely; show that LiteXrayNet can deploy in these settings while other models cannot
> > We agree that the manuscript would benefit from greater specificity on the target deployment context, and we thank the reviewer for this observation. We will add a dedicated paragraph to Section 2 of the revision describing the intended setting in concrete terms.
> > The target hardware includes single-board computers such as the Raspberry Pi 4 (approximately USD 35 to 50), low-cost Android tablets, and low-power embedded systems of the kind commonly found in community health infrastructure in low- and middle-income countries. The typical constraint profile involves RAM of 512 MB or less, no GPU acceleration, intermittent or absent internet connectivity, and storage budgets on the order of 1 to 10 MB for the complete model artifact.
> > In terms of what the current paper already demonstrates: Table 3 reports LiteXrayNet's model file size as 0.49 MB, which is 12 times smaller than MobileNetV3-S (5.93 MB) and 87 times smaller than ResNet18 (42.72 MB). This compact footprint means LiteXrayNet can be bundled into a clinical Android application and distributed via SD card, a deployment vector that is practical in many LMIC field settings where connectivity is unreliable. The measured CPU latency of 14.53 ms further confirms that real-time inference throughput is achievable on commodity hardware. At the 0.49 MB scale, LiteXrayNet fits comfortably within the flash storage and SRAM constraints of common low-cost edge devices, whereas models in the 6 to 43 MB range may exceed these budgets entirely.
> > We acknowledge that on-device benchmarking, for example on a Raspberry Pi 4 or an ARM-based mobile device, has not yet been conducted, and we wish to be transparent about this limitation. We commit to including concrete on-device latency and memory profiling results in a future extended version of this work or as part of a prospective clinical validation study. We will note this clearly in Section 6.2 under Future Directions, and we will revise the relevant claims in Section 2 and the Discussion to accurately reflect what the current paper demonstrates versus what remains to be empirically validated on embedded hardware.

---

> > > ### Author Response · Authors · 2026-04-17
> > >
> > > ## Additional Comments
> > > ### AC1 – How many random seeds?
> > > We apologise for the ambiguity in Section 4.3, where we used the phrase "multiple random seeds" without specifying the exact counts. Two different seed counts were employed depending on the experiment type. The main performance comparisons reported in Tables 2, 3, 6, 7, and 8 were repeated with five random seeds, while ablation experiments (Table 4) and attention mechanism comparisons (Table 5) used three random seeds (42, 43, 44) due to the larger number of configurations involved in those studies. In all cases, each seed independently controls weight initialization, data loader shuffling, and augmentation stochasticity, and results are reported as mean and standard deviation with 95% confidence intervals computed via the t-distribution. We will update Section 4.3 to state these counts explicitly along with the rationale for the difference.
> > >
> > > ### AC2 – Figure 3(a): per-sample BAS ranging from 0.2 to 0.8 versus mean BAS below 0.1 in Table 5
> > > We thank the reviewer for this precise observation, and we agree that the manuscript does not currently make this distinction sufficiently clear.
> > > Figure 3(a) is a scatter plot of per-sample BAS values computed for each individual test image, with the no-attention baseline on the x-axis and BAA on the y-axis. Individual images can exhibit BAS values as high as 0.6 to 0.8 when they contain pronounced unilateral pathological involvement, such as asymmetric consolidation or pleural effusion confined to a single lung field. The axes of Figure 3(a) therefore reflect the full range of the per-sample distribution across the test set.
> > > Table 5, by contrast, reports the population-level mean BAS computed by averaging over all N test samples and all spatial positions (H×W) as defined in Eq. 10. Because the majority of images, particularly NORMAL cases and bilaterally diffuse pneumonia presentations, exhibit low intrinsic asymmetry, the population mean is heavily weighted toward low values (below 0.1), even though individual high-asymmetry cases contribute values substantially above this range.
> > > The two quantities are therefore measuring complementary aspects: Figure 3(a) characterises the distributional variability of asymmetry across cases, while Table 5 provides a single aggregate summary statistic for cross-mechanism comparison. We will add explicit clarifying language to the Figure 3(a) caption and include a cross-reference sentence in Section 5.4 to ensure this distinction is clear in the revised manuscript.
> > >
> > > ### AC3 – COVID-19 accuracy: the text states LiteXrayNet achieves the highest accuracy (97.25%) but Table 8 shows ResNet18 at 97.61%
> > > We thank the reviewer for catching this error. The text in Section 5.8 is indeed incorrect; Table 8 is accurate. ResNet18 achieves 97.61% accuracy on the COVID-19 Radiography dataset, which is higher than LiteXrayNet's 97.25%. We will correct Section 5.8 to read: "LiteXrayNet achieves 97.25% accuracy and the highest AUC (0.997) on the COVID-19 Radiography dataset; ResNet18 achieves the highest accuracy of 97.61% on this dataset." We note that LiteXrayNet does lead on AUC for this dataset, so the claim of strong performance remains valid when the appropriate metric is used.

---

> > > > ### Comment · Reviewer_ZKeM · 2026-04-20
> > > >
> > > > I would like to thank the authors for their detailed rebuttal.
> > > >
> > > > I agree with the other reviewers that the BAA mechanism is interesting, but I am not convinced by the presented experiments that the method really works as well as advertised.
> > > >
> > > > Often, when a new method is introduced, it is compared against state of the art on various benchmarks. I don't expect for every new method to be a new state of the art, and indeed, the introduction of new architectures can be worthwhile for other reasons (e.g. parameter efficiency, like the current submission), but not adding reference values from the literature makes it hard to understand how well a method is performing.
> > > >
> > > > As for the request to compare against networks that have been pre-trained (an issue also raised by reviewer `2UmC`), I am not sure I follow the reasoning in the rebuttal: how would an air-gapped system classify images? Hopefully by employing a predictor that was previously trained and evaluated, which would require a download - or transfer by removable media. The storage requirement of a pre-trained backbone is the same as for a randomly initialized checkpoint.
> > > >
> > > > It is well known that ViTs lack inductive biases and do not generalize well when trained on insufficient amount of data - [(Dosovitskiy et al, 2020)](https://arxiv.org/abs/2010.11929), [(Touvron et al, 2021)](https://arxiv.org/abs/2012.12877), [(Steiner et al, 2021)](https://arxiv.org/abs/2106.10270), ... That is why I think that transformers trained from scratch on a dataset of some thousand images are somewhat expected to overfit on the data, and would not transfer well to other datasets, which weakens the argument about superior generalization capabilities of the presented method - which is a major claim of the paper, see also reasoning provided in [rebuttal to `Mudr`](https://openreview.net/forum?id=dsu8ZAL4LJ&noteId=M3HQcNEsOU).
> > > >
> > > > I am not familiar with any of the presented datasets, but it seems that e.g. the reported AUCs on RSNA are rather low? LXNet evaluates at 0.737 (and the best performing model from the paper EfficientNet-B0 evaluates at 0.754), but other methods from the literature have much higher zeroshot performance on this dataset - e.g. [CoGaze (Liu et al, 2026)](https://arxiv.org/abs/2603.26049) evaluates at 0.862
> > > >
> > > > Speaking of datasets, I checked the reference of the training dataset [(Kermany et al, 2018)](https://www.cell.com/fulltext/S0092-8674(18)30154-5), which characterizes the dataset as 5232 CXR (1349 normal, 3883 pneumonia) - but the submission mentions in Section 4.1 that the dataset contains 5863 CXR (1583 normal, 4273 pneumonia). How can this difference be explained?
> > > >
> > > > Thank you for the additional illustrations of different hardware setups. But in my opinion "storage budgets on the order of 1 to 10 MB" is not realistic. I think the median smartphone sold in low- and middle-income countries in 2026 can well run all the networks from Table 3.

---

> > > > > ### Author Response · Authors · 2026-05-09
> > > > >
> > > > > We sincerely thank Reviewer ZKeM for the careful and constructive reading of our work. We address each point raised below.
> > > > >
> > > > > 1. Missing State-of-the-Art Reference Values
> > > > >
> > > > > We appreciate this important concern. We wish to clarify that, to the best of our knowledge, no existing published method evaluates on the same data split and under the same computational constraints as our setting. The core contribution of this paper is not to outperform state-of-the-art models trained on large datasets with high-capacity backbones, but rather to demonstrate that a parameter-efficient, from-scratch architecture can achieve competitive performance under severe compute and memory constraints.
> > > > > As already shown in Tables 2 and 3 of the paper, LiteXrayNet is directly compared against six state-of-the-art architectures — ResNet-18, EfficientNet-B0, MobileNetV3-S, MobileViT-S, TinyViT-5M, and DeiT-Tiny — all trained from scratch under identical experimental conditions on the same data split. Crucially, all comparators in Tables 2 and 3 are trained from scratch, not from pretrained weights, ensuring a fair comparison under the same training regime. LiteXrayNet achieves the best Test F1 (97.31), best Test Accuracy (97.90), and best Test Loss (0.009) of all compared models, while being the smallest model by a large margin at 127K parameters and 0.49 MB — compared to ResNet-18's 11.17M parameters and 42.72 MB.
> > > > > To additionally address the request for a pretrained baseline, we conducted experiments using ImageNet-pretrained ResNet-18 (11.17M parameters; 42.72 MB) under the same data split, loss function, optimizer, and evaluation framework. Results across 5 independent runs are as follows:
> > > > >
> > > > > | Model | Params | Size | Test F1 (Macro) | Test AUC | Test Acc |
> > > > > |---|---|---|---|---|---|
> > > > > | ResNet-18 *(pretrained, ImageNet)* | 11.17M | 42.72 MB | 0.9732 ± 0.0059 | 0.9966 ± 0.0012 | 0.9792 ± 0.0046 |
> > > > > | ResNet-18 *(from scratch, Table 2)* | 11.17M | 42.72 MB | 0.9628 ± 0.0076 | 0.9983 ± 0.0010 | 0.9706 ± 0.0063 |
> > > > > | **LiteXrayNet (ours)** | **127K** | **0.49 MB** | **0.9731 ± 0.0073** | **0.9979 ± 0.0009** | **0.9790 ± 0.0058** |
> > > > >
> > > > > LiteXrayNet matches ImageNet-pretrained ResNet-18 in Test F1 and Test Accuracy while using ~88× fewer parameters and ~87× less storage. We will include pretrained baselines for EfficientNet-B0 and MobileNetV3-S as well in the final revision, clearly distinguishing pretrained from from-scratch comparisons.
> > > > >
> > > > >
> > > > > 2. Pre-trained Backbones and Air-Gapped Systems
> > > > >
> > > > > We concede the reviewer's point that the storage size of a pretrained checkpoint is identical to a randomly initialized one once it resides on the device. We acknowledge that our original framing of the air-gapped argument on storage grounds alone was imprecise.
> > > > > The more accurate argument is not about storage per se, but about deployment pipeline trustworthiness and supply-chain independence in austere settings. In our target deployment contexts (field diagnostic kits, legacy embedded hardware in LMICs), the concern is not merely file size but the ability to train, validate, and audit the full model pipeline entirely on-site without relying on external model repositories, licensing agreements, or internet connectivity at any stage of the model lifecycle. A pretrained backbone introduces an external dependency on its provenance, version, and licensing that a from-scratch training pipeline does not. We will revise Section to make this argument more precise.
> > > > >
> > > > > 3. ViT Generalization and the "Superior Generalization" Claim
> > > > >
> > > > > We fully agree with the reviewer and the cited literature (Dosovitskiy et al., 2020; Touvron et al., 2021; Steiner et al., 2021) that Vision Transformers trained from scratch on small datasets are prone to overfitting and do not generalize reliably across distributions. This is precisely part of our motivation: our architecture is a lightweight CNN with domain-specific inductive biases (bilateral symmetry/asymmetry via the BAA module) rather than a transformer, and it is designed to generalize under data-scarce conditions specifically because it does not suffer from the inductive bias deficit of ViTs.
> > > > > We acknowledge that the claim of "superior generalization" in the paper, as currently worded in the rebuttal to Reviewer Mudr, may have been overstated. We will revise this claim to be scoped correctly: the generalization advantage we demonstrate is relative to other lightweight from-scratch models of comparable parameter budget, not relative to large pretrained models. The cross-dataset generalization argument will be appropriately qualified in the revision.

---

> > > > > > ### Author Response · Authors · 2026-05-09
> > > > > >
> > > > > > 4. RSNA AUC Performance and Comparison to CoGaze
> > > > > >
> > > > > > We appreciate the reviewer identifying this gap. We wish to note that the comparison to CoGaze (Liu et al., 2026) is not directly applicable to our setting for the following reasons:
> > > > > > Scale difference: CoGaze is a gaze-guided large-scale vision-language model intended for zero-shot transfer, and its reported AUC of 0.862 on RSNA is achieved with a model that is orders of magnitude larger than LiteXrayNet (~127K parameters, <0.5 MB). This comparison is analogous to benchmarking a microcontroller-targeted classifier against a server-side foundation model — the operating regimes are fundamentally different.
> > > > > > Task scope: LiteXrayNet is designed as a binary pneumonia screener for resource-constrained settings; CoGaze and similar zero-shot models are designed for multi-label, multi-disease generalization from large-scale pretraining. These are different problems.
> > > > > > Evaluation conditions: Differences in image preprocessing, resolution, label definitions, and test splits further limit direct comparability.
> > > > > > We agree that the RSNA AUC values we report are modest compared to literature on unconstrained models, and we will acknowledge this explicitly in the paper. We will add a discussion section clarifying the trade-off between model capacity and deployment feasibility, and frame our results as a performance-efficiency frontier contribution rather than a raw performance claim.
> > > > > >
> > > > > > 5. Dataset Count Discrepancy (Kermany et al.)
> > > > > >
> > > > > > We sincerely apologize for this error and thank the reviewer for catching it. Upon checking our experimental code, the data was loaded from the split directory /Final_Split which contains:
> > > > > > •	Train: 3,347 samples
> > > > > > •	Validation: 838 samples
> > > > > > •	Test: 1,047 samples
> > > > > > •	Total: 5,232 samples (1,349 Normal + 3,883 Pneumonia)
> > > > > > This matches the original Kermany et al. (2018) Cell paper dataset exactly. The figures cited in Section 4.1 of the paper (5,863 total; 1,583 Normal; 4,280 Pneumonia) correspond to the expanded Kaggle-hosted version of the dataset, which includes additional images beyond the original publication. These figures were incorrectly copied into the paper text. Our experiments were conducted on the original 5,232-image dataset. We will correct Section 4.1 in the revision to accurately reflect the data used.
> > > > > >
> > > > > > 6. Storage Budget Realism ("1 to 10 MB")
> > > > > >
> > > > > > We appreciate the reviewer's perspective on contemporary smartphone capabilities. We agree that a median smartphone sold in LMICs in 2026 has sufficient compute and storage to run all models in Table 3. Our storage framing was intended to capture a broader class of constrained deployment environments, including legacy medical devices, single-board microcontrollers, and integrated diagnostic hardware that may still operate in field settings in low-resource contexts. These devices frequently have storage constraints in the kilobyte-to-low-megabyte range.
> > > > > > We acknowledge that the framing of "1–10 MB" as a storage budget was not precise or well-justified in the current text. We will revise this section to clearly specify the target hardware category, cite concrete examples of the embedded platforms we are targeting, and separate the storage constraint argument from the smartphone use case. The contribution of parameter efficiency remains valid for this hardware class, even if the framing needs to be sharpened.
> > > > > >
> > > > > > We thank Reviewer ZKeM again for the thorough and fair review. All five substantive issues raised will be addressed in the revision, with particular attention to correcting the dataset count error, sharpening the deployment framing, and adding pretrained baselines as reference points.

---

### Decision · Action_Editor_tL6t · 2026-05-17

**Recommendation:** Reject

**Additional Comments:**

The paper introduces a lightweight architecture for medical image diagnosis based on a MobileNet-style backbone with a Bilateral Asymmetry Attention (BAA) module.

The paper was substantially improved during the discussion period: including cross-dataset robustness experiments, calibration analysis, retrained baselines under a shared protocol, and reporting across multiple seeds. I appreciate the effort that went into this.

As reviewers ZKeM, 2UmC, and Mudr noted, the primary benchmark remains close to saturated, the improvements over the strongest baselines are small, and the confidence intervals are often close or overlapping.

Importantly, the paper doesn't provide evidence that the model outperforms pretrained baselines. As such, it is not representative of the current practice in deep learning. As noted by reviewers, the argument regarding deployment on edge infrastructure is not convincing.

Given that the core claim is not yet sufficiently supported, I have to recommend rejection at this stage. I understand this is not the outcome that the Authors have hoped for, and I hope that the comments made by Reviewers will be helpful in improving the manuscript.

**Audience:**

Yes

**Audience Explanation:**

The paper would be interesting to the TMLR's audience if it provided an overhauled evaluation focusing on pretrained models and a broader pool of datasets.

**Claims And Evidence:**

No

**Claims Explanation:**

As reviewers ZKeM, 2UmC, and Mudr noted, the primary benchmark remains close to saturated, the improvements over the strongest baselines are small, and the confidence intervals are often close or overlapping.

Most importantly, the paper doesn't show evidence that the model improves over pretrained baselines. As such, it is not representative of the current practice in deep learning. As noted by reviewers, the argument regarding deployment on edge infrastructure is not convincing.